# Scaling Laws for Multilingual Language Models

## Abstract

We propose a novel scaling law for general-purpose decoder-only language models (LMs) trained on multilingual data, tackling the problem of balancing languages during multilingual pretraining. A primary challenge in studying multilingual scaling is the difficulty of analyzing individual language performance due to cross-lingual transfer. To address this, we shift the focus from individual languages to language families. We introduce and validate a hypothesis that the test cross-entropy loss for each language family is determined solely by its own sampling ratio, independent of other languages in the mixture. This insight simplifies the complexity of multilingual scaling and make the analysis scalable to an arbitrary number of languages. Building on this hypothesis, we derive a power-law relationship that links performance with dataset size, model size and sampling ratios. This relationship enables us to predict performance across various combinations of the above three quantities, and derive the *optimal sampling ratios* at different model scales. To demonstrate the effectiveness and accuracy of our proposed scaling law, we perform a large-scale empirical study, training more than 100 models on 23 languages spanning 5 language families. Our experiments show that the optimal sampling ratios derived from small models (85M parameters) generalize effectively to models that are several orders of magnitude larger (1.2B parameters), offering a resource-efficient approach for multilingual LM training at scale.

## 1 Introduction

Scaling has proven to be a powerful strategy for improving the performance of language models (LMs) across a range of tasks (Thoppilan et al., 2022; Smith et al., 2022; Achiam et al., 2023). Due to the enormous cost associated with training larger models (Rae et al., 2021; Dubey et al., 2024), *neural scaling laws* (Kaplan et al., 2020; Henighan et al., 2020; Hoffmann et al., 2022; Krajewski et al., 2024) have emerged to be an effective approach to a priori quantify and predict the gains from scaling up model size, dataset size, and computational resources. Previous works on scaling laws predominantly focus on monolingual LMs, neglecting the increasingly crucial role of *multilinguality* for LMs to cater to diverse linguistic populations and global users (Conneau & Lample, 2019; Conneau et al., 2020; Yang et al., 2024; Dubey et al., 2024). With increasing emphasis on inclusion and wider language support (Qin et al., 2024), there is a pressing need to extend these scaling laws to multilingual LMs to address unique challenges, particularly how to balance training across different languages effectively.

Although there exist some studies that examine multilingual scaling laws (Gordon et al., 2021; Ghorbani et al., 2022; Fernandes et al., 2023; Chen et al., 2024), they often focus on the specific problem of neural machine translation (NMT). These studies predominantly utilize encoder-decoder transformer architectures, which are not widely applied to generation tasks. Furthermore, they typically restrict their analysis to bilingual settings without capturing the complexities of language interactions. This omission is critical, as *cross-lingual transfer*, a key benefit in multilingual training, plays a significant role in improving performance across languages (Arivazhagan et al., 2019; Xian et al., 2022; Patra et al., 2023).

To address the aforementioned issues, in this work, we focus on building scaling laws for *general purpose decoder-only* LMs pretrained on *multilingual data*. Our contributions are grounded in a realistic and novel hypothesis that unlocks a broader understanding of multilingual scaling, making our analysis scalable to *an arbitrary number of languages*.

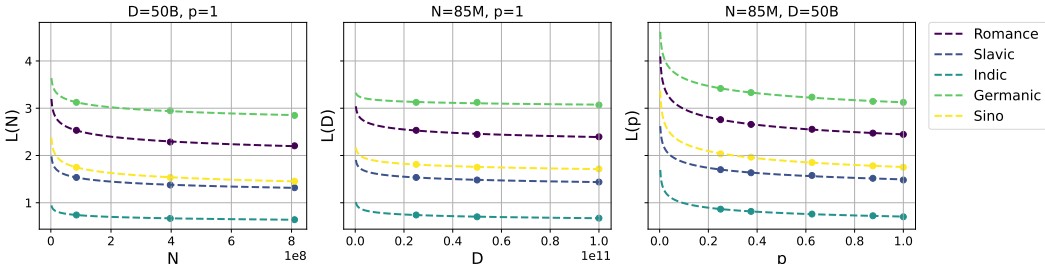

Figure 1: We propose a multilingual scaling law connecting the test cross-entropy loss ($\mathcal{L}$) with model size in number of parameters ($N$), dataset size ($D$) and sampling ratios for different language families ($\boldsymbol{p}$). The plots illustrate a power-law relationship by varying one quantity while fixing the other two for five language families.

To substantiate our claims, we conduct a large-scale empirical study by training more than 100 LMs covering 23 languages spanning 5 language families. Through this study, we propose a new multilingual scaling law that significantly enhances the predictive power for multilingual LM performance. This law provides a succinct power-law relationship between the test cross-entropy loss, model size, dataset size, and the sampling ratios of different language families (shown in Figure 1). The scaling law enables us to derive an accurate performance prediction across a wide range of combinations of the three quantities. More importantly, with this predictive power, we can directly derive *the optimal sampling ratios* for language families in the training mixture *across varied model and dataset sizes* by only training small models.

In summary, our key contributions are:

- **Novel hypothesis for cross-lingual transfer**: One major challenge in analyzing multilingual LMs is the inability to isolate performance for each language due to cross-lingual transfer, where the performance of one language depends on other linguistically related languages trained jointly. Our key insight in tackling this problem is a hypothesis that the performance of each *language family* is independent of other language families in the training mixture. We empirically verify this hypothesis, which enables us to directly analyze the relationship between a language family's performance and its sampling ratio. Specifically, we show that the test cross-entropy loss of each language family depends primarily on its own sampling ratio, independent of the sampling ratios of other language families in the training mixture. This provides an important simplification for analyzing multilingual scaling behavior.

- **Multilingual scaling law**: Based on the validated hypothesis, we propose a scaling law that relates the test cross-entropy loss ($\mathcal{L}_i$) for each language family $i$ to model size ($N$), dataset size ($D$) and language family sampling ratios ($\boldsymbol{p}$):

$$\mathcal{L}_i(N, D, \boldsymbol{p}) = \left( E_i + \frac{A_i}{N^{\alpha_i}} + \frac{B_i}{D^{\beta_i}} \right) p_i^{-\gamma_i},$$

  where $E_i, A_i, B_i, \alpha_i, \beta_i, \gamma_i$ are fixed parameters for the $i$-th family. One key implication of the above form is that the scaling law of each language family only depends on its own sampling ratio $p_i$, independent of the sampling ratios of other families $p_{j \neq i}$. Additionally, we discover that the exponent $\gamma_i$, which governs how much loss reduces as the proportion of data from family $i$ increases, remains invariant to model size $N$ and dataset size $D$. This finding further strengthens the applicability of our scaling law across different compute scales.

- **Derivation of the optimal sampling ratios**: Leveraging the proposed scaling law, we derive *the optimal sampling ratios* that minimize the total loss for the LM, thus providing an effective data mixing strategy for multilingual pretraining. We validate the optimality of these ratios by comparing them against other baseline sampling methods. We demonstrate that the optimal sampling ratios obtained from small models (85M parameters) generalize well to models that are several orders of magnitude larger (1.2B parameters). This insight implies that for resource-efficient LM training, practitioners can optimize training mixtures for large-scale models by only training smaller and more affordable models, drastically reducing computational overhead while maintaining performance consistency across model scales.

Table 1: List of 23 languages in our study with division into 5 language families.

| Families | Languages |
|---|---|
| Germanic | English, German, Dutch, Danish |
| Romance | Spanish, French, Italian, Romanian, Catalan |
| Slavic | Russian, Ukrainian, Slovak, Serbian, Croatian |
| Indic | Hindi, Bengali, Nepali, Marathi, Tamil, Telugu, Kannada, Malayalam |
| Sino-Tibetan | Chinese |

## 2 EXPERIMENTAL SETUP

**Model.** We train decoder-only Transformer models (Vaswani, 2017) in four sizes, ranging from 85M to 1.2B non-embedding parameters. The model sizes are determined by adjusting the number of layers, hidden sizes, and the number of attention heads. We detail the model configurations in Appendix B.

**Data.** We focus on the pretraining stage of multilingual language models. We use the CommonCrawl dataset (Conneau et al., 2020; Wenzek et al., 2020). Following the approach of Lai et al. (2023), we select 23 languages based on diversity and representativeness among the total 100 languages. We follow common practice (Fan et al., 2021; Costa-jussà et al., 2022) to group the languages into five language families based on linguistic similarities: Romance, Slavic, Indic, Germanic and Sino-Tibetan. The detailed languages within each family are presented in Table 1. We use the `cl100k_base` tokenizer[1] to tokenize the corpus. For each language in the training corpus, we apply a 90/10 random split, where the latter 10% is held-out for test cross-entropy loss evaluation. More details about the dataset can be found in Appendix A.

## 3 THE MULTILINGUAL SCALING LAW

We study the relationship between performance, model sizes, dataset sizes and sampling ratios. In Section 3.1, we begin by motivating the problem of multilingual scaling laws. In Section 3.2, we formulate and validate a hypothesis to address this problem in a tractable manner. In Section 3.3, we propose a power-law relationship between the performance and sampling ratios. In Section 3.4, we incorporate model size and dataset size into this relationship to establish a comprehensive scaling law. Finally in Section 3.5, we validate the fitting of our scaling law.

### 3.1 PROBLEM STATEMENT

During the pretraining stage of an LM, given $n$ languages in the training data mixture, we explore the relationship between the total test cross-entropy loss $\mathcal{L}$, model size $N$, dataset size $D$ (the total token count) and sampling ratios of each languages $\boldsymbol{p} = [p_1, \cdots, p_n]$ in the training data mixture. Here, $\boldsymbol{p}$ is a probability vector, i.e., $\boldsymbol{p} \in \Delta_n$, where $\Delta_n$ is the $(n-1)$-dimensional probability simplex. Specifically, we want to fit the relationship

$$\mathcal{L}(N, D, \boldsymbol{p}) = \sum_{i=1}^{n} w_i \mathcal{L}_i(N, D, \boldsymbol{p}), \tag{1}$$

where $\mathcal{L}_i$ denotes the test cross-entropy loss for language $i$, and $w_i{}^2$ represents the *user-defined* preference for each language, indicating its importance. For instance, one can emphasize a particular language by increasing the corresponding $w_i$. Note that when $w_i = p_i$, this loss reflects the total empirical loss. This relationship is informative and predictive, as it directly leads to the following key capabilities:

- **Performance prediction**: The relationship allows us to predict performance of LMs trained on unseen sampling ratios. We can plug any combinations of $N$, $D$ and $\boldsymbol{p}$ into Eq. 1 to obtain the loss without conducting the training.

---

[1] https://github.com/openai/tiktoken

[2] The weights should be non-negative. To get the optimal sampling ratios, only the ratios between $w_i$ matter.

- **Optimal sampling ratios**: The framework provides a mechanism to determine the optimal sampling ratios $p$ leading to minimal total losses given $N$ and $D$. This is achieved by solving the following optimization problem:

$$p_w^\star = \arg\min_{p \in \Delta_n} \sum_{i=1}^n w_i \mathcal{L}_i(N, D, p).$$

More importantly, we will demonstrate that the $p_w^\star$ obtained from small models remains near optimal on significantly larger models.

## 3.2 HYPOTHESIS

For simplicity, we first consider a setting where model size and dataset size are fixed, and we only study the relationship between losses and sampling ratios, i.e., $\mathcal{L}_i(p)$. Fitting each individual $\mathcal{L}_i(p)$ directly is intractable as $p$ is an $n$-dimensional vector, and it is computationally infeasible to sample sufficiently many $p$ to effectively cover the $n$-dimensional space. To address this challenge, we propose and validate a realistic hypothesis to reduce this complexity.

Despite previous attempts to study the relationship between performance and sampling ratios (Fernandes et al., 2023; Chen et al., 2024), a key factor often overlooked is the impact of *cross-lingual transfer*. In particular, previous works directly assume $\mathcal{L}_i(p) = \mathcal{L}_i(p_i)$, implying that the loss of each language depends solely on its own sampling ratio, regardless of the combination of other jointly trained languages. However, this assumption is equivalent to the statement that there exists no knowledge transfer across languages, which does not hold true in general. For instance, due to insufficient training data, low-resource languages (e.g., Catalan) benefit from knowledge transfer from linguistically similar high-resource languages (e.g., Spanish) (Arivazhagan et al., 2019).

To tackle the problem, we provide a more realistic hypothesis. Instead of studying the sampling ratios of each individual language, we focus on language groups with the following properties: **i) Minimal cross-group transfer**: The majority of cross-lingual transfer occurs within a group, with minimal transfer across groups. **ii) Data sufficiency**: Each group provides a substantial amount of data, reducing the need to model dynamics of low-resource languages. We find *language families* to be an intuitive and effective grouping, as they are defined based on linguistic similarities, and each language family contains multiple languages, mitigating the low-resource issues associated with individual languages. [3] Then, we have the following hypothesis.

**Hypothesis 1** *During the pretraining of a multilingual language model, the test cross-entropy loss of each language family only depends on its own sampling ratio, regardless of the sampling ratios of other languages jointly trained.*

**Hypothesis testing.** We use a controlled experiment to test our hypothesis. We train an LM on a data mixture containing three language families: Romance, Germanic and Slavic. In the training, we fix the sampling ratio of Romance to be either 0.2 or 0.5, and vary the ratios of the other two families. At the individual language level, one might expect high-resource languages like English (within the Germanic family) to transfer knowledge to related languages in the Romance family (e.g., Spanish, French). However, our hypothesis focuses on *cross-family transfer* instead of cross-lingual transfer among individual languages. In Figure 2 (left), we observe that with a fixed sampling ratio, the Romance loss does not noticeably change regardless of the combination ratios of Germanic and Slavic data. This stability indicates that the primary source of performance variation is the sampling ratio of the family itself, rather than cross-family interactions. This finding supports our hypothesis that cross-family transfer is minimal, making it feasible to model each language family's loss as a function of its own sampling ratio. To strengthen our findings, we conduct the same experiment with three additional family combinations, detailed in Appendix C.

To further demonstrate that grouping by language families is an effective approach, we perform a similar experiment with *random groupings*. In this setup, both Group 1 and Group 2 contain languages from the Indic family. If our hypothesis about cross-family transfer holds, we would expect substantial transfer between these groups, since they share linguistic characteristics. As shown

---

[3]While language families are a natural grouping, other groupings based on criteria such as lexical overlap can also be considered, as long as they satisfy the two criteria.

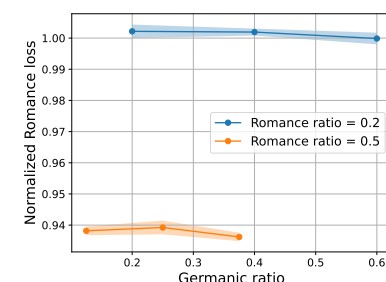 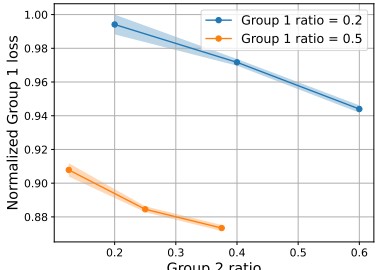

Figure 2: **Left:** The Romance loss remains stable as the Germanic sampling ratio varies, indicating minimal cross-family transfer and supporting our hypothesis that each language family's performance is primarily influenced by its own sampling ratio. **Right:** In contrast, when both groups contain Indic languages, Group 1 loss decreases as Group 2 sampling ratio increases, demonstrating significant cross-group transfer. This underscores the importance of grouping by language families for accurate analysis. The loss values are normalized by the mean loss at $p = 0.2$ to align the plot scales. Shaded areas indicate standard deviation.

in Figure 2 (right), the performance of Group 1 improves with increasing sampling ratio of Group 2, which indicates significant cross-group transfer when languages from the same family are split into separate groups, highlighting the importance of proper family-based grouping. We include a comparison of the trajectory of losses during training in Appendix C to further validate the claims.

These experimental results validate our hypothesis, which enables a convenient simplification of Eq. 1: For each language family $\text{fam}_i$, we have

$$\mathcal{L}_{\text{fam}_i}(\boldsymbol{p}) = \mathcal{L}_{\text{fam}_i}(p_{\text{fam}_i}),$$

where we can safely focus on the sampling ratio for each language family, rather than accounting for interactions across different families. Subsequently, for the ease of presentation, we omit the subscript "fam" and directly use $\mathcal{L}_i$ and $p_i$ to represent the loss and sampling ratio for the $i$-th *language family*.

### 3.3 FITTING THE RELATIONSHIP: $\mathcal{L}_i(p_i)$

In this section, we first study the relationship between the loss and sampling ratios given fixed model size and dataset size. The loss of a language family $\mathcal{L}_i$ can be modeled by a power-law relationship:

$$\mathcal{L}_i(p_i) = \mathcal{L}_i^{\star} \cdot p_i^{-\gamma_i}, \tag{2}$$

where both $\mathcal{L}_i^{\star}$ and $\gamma_i$ are fixed parameters.

The choice of power-law formulation follows naturally from previous studies on scaling laws, which demonstrate that the loss exhibits a power-law behavior with respect to dataset size (Kaplan et al., 2020). Our proposed form extends this concept to the multilingual setting, where the sampling ratio $p_i$ can be viewed as analogous to relative dataset size for each family. The power-law form captures the intuitive notion that increasing the sampling ratio $p_i$ leads to diminishing returns in terms of reducing the test cross-entropy loss. In other words, as the amount of data for a language family increases, its marginal contribution to performance improvement decreases.

**Interpretation of parameters.** $\mathcal{L}_i^{\star}$ represents the test loss when a language family $i$ constitutes the entire training dataset ($p_i = 1$). It indicates the baseline difficulty of modeling this family alone. As the model size $N$ increases, $\mathcal{L}_i^{\star}$ typically decreases due to higher model capacity.

On the other hand, $\gamma_i$ indicates the decay rate of loss for the $i$-th family given increasing proportion of the $i$-th family in the training data mixture. In general, a larger $\gamma_i$ indicates that the language family benefits more from increasing sampling ratio and should be prioritized in the training mixture for an overall performance improvement.

**Empirical validation.** To empirically validate the power-law relationship, we train LMs with 397M non-embedding parameters across various data mixtures to obtain losses on 5 sampling ratios ($p_i \in \{0.25, 0.375, 0.625, 0.875, 1\}$) for each language family. An important implication of the language family independence hypothesis is that a single training run generates data points for all involved language families. For example, training a model with mixture such as ($p_{Romance} =$

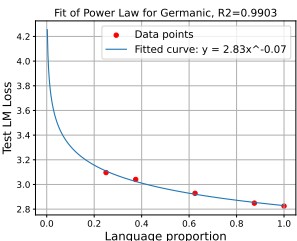 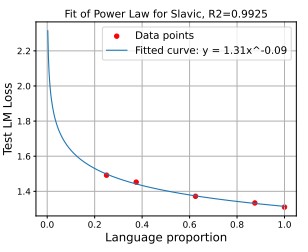

Table 2: Fitted parameters for different language families for 397M model size and 50B token count.

| Language | $\mathcal{L}_i^\star$ | $\gamma_i$ |
|---|---|---|
| Romance | 2.186 | 0.080 |
| Slavic | 1.314 | 0.094 |
| Indic | 0.635 | 0.131 |
| Germanic | 2.829 | 0.068 |
| Sino-Tibetan | 1.557 | 0.109 |

Figure 3: Fitting for Germanic and Slavic families with 50B tokens. The high R-squared values indicate an accurate fit of the power-law relationship.

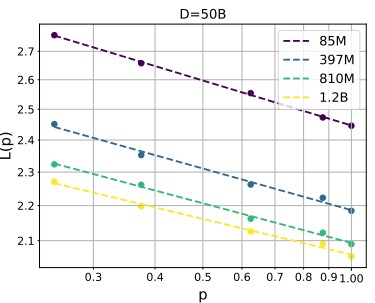 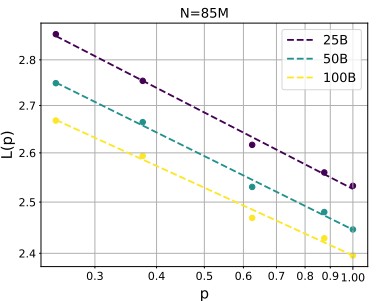

Figure 4: **Left:** For a fixed token count $D$, there is a linear relationship between $\log(\mathcal{L}_i)$ and $\log(p)$ for different values of model size $N$. **Right:** For a fixed model size $N$, there is a linear relationship between $\log(\mathcal{L}_i)$ and $\log(p)$ for different values of dataset size $D$. The parallel lines indicate that the decay rate $\gamma_i$ does not depend on either $N$ or $D$. Both axes are in log-scale.

$0.375, p_{Germanic} = 0.625$) provides us with 2 data points, one for each language family. By strategically using such bilingual models, we only need to train 15 runs instead of naively running 25 experiments to gather datapoints required for fitting all 5 language families across all 5 different $p_i$ values.

Figure 3 demonstrates the results of this fitting, and the high R-squared values indicate an accurate fitting for each language family. The results for other families with different model sizes are similar in terms of R-squared, and are deferred to Appendix D.

### 3.4 FITTING THE JOINT RELATIONSHIP: $\mathcal{L}_i(N, D, p)$

Building on the results from the previous section, we extend the relationship by incorporating both model size $N$ and dataset size $D$. We achieve this by expanding $\mathcal{L}_i^\star$ (the mono-family loss) from Section 3.3, as it depends on both $N$ and $D$. Considering that $\gamma_i$ could also depend on $N$ and $D$, we rewrite Eq. 2 into

$$\mathcal{L}_i(N, D, p_i) = \mathcal{L}_i^\star(N, D) \cdot p_i^{-\gamma_i(N,D)}. \tag{3}$$

First, we show that $\gamma_i$ is independent of both $N$ and $D$ by examining the equation in the log scale:

$$\log(\mathcal{L}_i(N, D, p_i)) = \log(\mathcal{L}_i^\star(N, D)) - \gamma_i(N, D) \log(p_i).$$

Intuitively, when varying $N$ and $D$, if the slope remains constant, then $\gamma_i$ is independent of them. In Figure 4, we observe that given a fixed $D$, $\gamma_i(N, D)$ does not change as $N$ varies, confirming its independence from $N$. Similarly, the slope remains constant when $N$ is fixed and $D$ changes, verifying that $\gamma_i(N, D)$ is also independent of $D$. Thus, we can simplify Eq. 3 into

$$\mathcal{L}_i(N, D, p_i) = \mathcal{L}_i^\star(N, D) \cdot p_i^{-\gamma_i}. \tag{4}$$

Next, we investigate the form of $\mathcal{L}_i^\star(N, D)$, which represents the mono-family performance as a function of model size $N$ and dataset size $D$. To do this, we leverage existing monolingual scaling

results from Hoffmann et al. (2022):

$$\mathcal{L}(N, D) = E + \frac{A}{N^\alpha} + \frac{B}{D^\beta}, \tag{5}$$

where $E, A, B, \alpha, \beta$ are fixed parameters. The functional form of this law is universal and applicable to any dataset, and the specific values of these parameters are dataset dependent. Thus, we directly apply it to Eq. 4, and arrive at the following joint law:

$$\mathcal{L}_i(N, D, \boldsymbol{p}) = \left( E_i + \frac{A_i}{N^{\alpha_i}} + \frac{B_i}{D^{\beta_i}} \right) p_i^{-\gamma_i}, \tag{6}$$

where $E_i, A_i, B_i, \alpha_i, \beta_i, \gamma_i$ are fixed parameters specific to each language family, all of which are *independent* of $N$ and $D$. The justification of this functional form can be found in Appendix E.

**Comparison with previous works.** The closest prior work to ours (Fernandes et al., 2023) introduce a multilingual scaling law in neural machine translation (NMT). They study a *bilingual* setting, translating English into Chinese or German. Their scaling law is expressed as:

$$\mathcal{L}_i(N, p) = \beta_{p,i} N^{-a_i} + \mathcal{L}_\infty^{(i)}, \tag{7}$$

where $\mathcal{L}_\infty^{(i)}$ and $a_i$ are fixed parameters, $\beta_{p,i}$ is a parameter *dependent* on the sampling ratio $p$, and $p$ is a *scalar* instead of a probability vector as in our setting. While this approach captures the effect of model size on bilingual translation tasks, our formulation offers several key improvements and broadens the scope in multiple dimensions:

- **Generalized framework**: Our scaling law applies to general-purpose language modeling tasks, whereas their work is focused on a specialized NMT setting with encoder-decoder architectures. The encoder-decoder architecture has limited use cases outside of NMT, limiting the applicability of their findings. In contrast, our scaling law is relevant to a broader spectrum of language modeling tasks, which is crucial given the prevalence of decoder only language models in recent times (Achiam et al., 2023; Jiang et al., 2023; Dubey et al., 2024; Yang et al., 2024; Abdin et al., 2024)

- **Multilingual scalability**: We significantly extend the scope to 23 languages across 5 language families, whereas Fernandes et al. (2023) focus on a bilingual setting. Our proposed law extends naturally to an arbitrary number of languages by considering cross-lingual transfer. In contrast, Eq. 7 does not generalize for more than 2 languages, as it lacks a mechanism for incorporating multilingual interactions.

- **Incorporation of Dataset Size**: Our scaling law explicitly incorporates the effect of dataset size, enabling a joint analysis of how both model size and data quantity impact performance. This is not considered in Eq. 7, limiting its ability to account for the full range of factors affecting multilingual model performance.

- **Simpler and more predictive form:** In Eq. 7, the parameter $\beta_{p,i}$ is dependent on the sampling ratio $p$ itself, making the equation not predictive for new sampling ratios. For unseen values of $p$, additional heuristics or retraining would be required to determine the corresponding $\beta_{p,i}$. In contrast, our proposed law decouples the dependency on $p_i$ through the power-law exponent $\gamma_i$, which remains constant across different sampling ratios. This makes our model more straightforward and fully predictive without requiring extra information for new values of $p_i$.

Overall, our proposed scaling law offers a more versatile and comprehensive framework for multilingual and general language modeling.

### 3.5 FITTING THE PARAMETRIC SCALING LAW

Subsequently, we fit parameters in Eq. 6 to describe the multilingual scaling. To estimate the parameters, we deploy a similar strategy as Hoffmann et al. (2022) to use the Huber loss[4] ($\delta = 0.001$) (Huber, 1964), as it is robust to outliers. The estimation is done by the BFGS algorithm (Nocedal, 1980). We present the fitting in Figure 5, where the fitted curves highlight that our proposed power law captures the relationship between loss, model size, dataset size and sampling ratios well. Furthermore, the right panel shows that our scaling law accurately predicts performance. These results confirm the effectiveness of our scaling law across. Fitting results for additional families can be found in Appendix D.2.

---

[4] Huber loss is only used in optimization of parameter fitting, which is different from the cross-entropy loss $\mathcal{L}$.

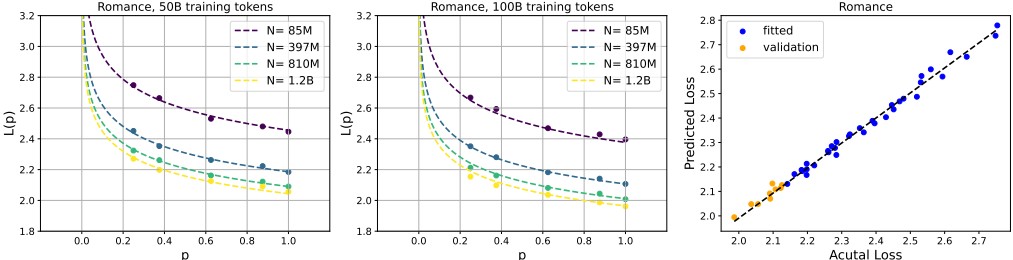

Figure 5: **Left & Middle:** Fitted law on 50B and 100B training tokens, showing that the scaling law well captures the relationship between loss, model size, dataset size and sampling ratios. **Right:** Predicted vs. actual losses with our scaling law. The fitting uses the top $80\%$ of the loss data (blue points) and then validated on the lower $20\%$ (orange points). The strong alignment between the predicted and actual losses demonstrates the predictive accuracy of the scaling law.

## 4    OPTIMAL ALLOCATION OF MULTILINGUAL SAMPLING RATIOS

With the general form of the total loss established, we proceed to compute the optimal data mixture defined by $\boldsymbol{p}$ by solving the following optimization problem:

$$\underset{\boldsymbol{p}\in\Delta_n}{\text{minimize}} \quad \mathcal{L}(\boldsymbol{p}) = \sum_i w_i \mathcal{L}_i(p_i) = \sum_i w_i \mathcal{L}_i^\star p_i^{-\gamma_i}, \tag{8}$$

where $w_i$ represents the user-defined preference of language family $i$. This formulation allows us to find the optimal sampling ratios $\boldsymbol{p}_{\boldsymbol{w}}^\star$ that minimize the total loss, taking into account the importance of each language family as specified by the preference vector $\boldsymbol{w}$.

**Analytical (approximate) solution.** The optimization problem can be solved analytically using the Lagrange multipliers method. Under the assumption $\gamma_i \ll 1$[5] we can obtain the approximate optimal $p_i^\star$ as (details presented in Appendix F)

$$p_i^\star \approx \frac{w_i \mathcal{L}_i^\star \gamma_i}{\sum_{i=1}^n w_i \mathcal{L}_i^\star \gamma_i}.$$

The solution shows that the optimal sampling ratios only depend on the products $w_i \mathcal{L}_i^\star \gamma_i$, meaning that the combination of the power law exponent and constant fully determines the ratios. Furthermore, if we use $w_i = 1/\mathcal{L}_i^\star$, the optimal ratios do not depend on $N, D$. In this case, the approximate solution sheds even more insight, as the optimal ratios depend only on the relative ratio of $\gamma_i / \sum_{i=1}^n \gamma_i$.

**Numerical solution.** Alternatively, one can directly use off-the-shelf numerical solver such as `scipy.optimize` to solve the optimization problem. We demonstrate that both methods result in similar optimal sampling ratios in Appendix F.

### 4.1    EXPERIMENTAL SETUP

**Preference vector $\boldsymbol{w}$.** We use two common choices of the preference vector: **i) Unweighted sum:** All $w_i = 1$ indicating equal weight for all languages with no specific preference. **ii) Normalized sum**: $w_i = 1/\mathcal{L}_i(N, D, 1) = 1/\mathcal{L}_i^\star(N, D)$. This is equivalent to a normalized sum of losses, where the loss of each language family is normalized by its mono-family performance. This approach compensates for differences in loss scales due to tokenizer vocabulary imbalances across languages (as seen in the varying $\mathcal{L}_i^\star$ values in Table 2). Normalizing by this loss balances the training across language families. This normalization technique is often utilized in the literature of multi-task learning, where the losses for different tasks vary in scales (Chen et al., 2018; He et al., 2024).

**Baselines.** We compare with three baseline weighting strategies: **i) Uniform sampling**: Each language family has equal sampling ratio, i.e., $p_i = 1/n$. **ii) Proportional to token count**: The sampling ratio is proportional to the token count of each family i.e., $p_i = D_i/D$, where $D_i$ is the token count for the $i$-th family, and $D$ is the total token count. **iii) Smoothed sampling** Conneau &

---

[5]Table 2 shows that $\gamma_i < 0.15$, so this is not an unreasonable assumption

Table 3: Sampling ratios and the resulting test cross-entropy loss for the 85M multilingual LM trained on 5 language families, with the total loss computed by the unweighted sum.

| | $p_{Ro}$ | $p_{Sl}$ | $p_{In}$ | $p_{Ge}$ | $p_{Si}$ | $\mathcal{L}_{Ro}$ | $\mathcal{L}_{Sl}$ | $\mathcal{L}_{In}$ | $\mathcal{L}_{Ge}$ | $\mathcal{L}_{Si}$ | Total loss |
|---|---|---|---|---|---|---|---|---|---|---|---|
| Uniform | 0.200 | 0.200 | 0.200 | 0.200 | 0.200 | 2.802 | 1.747 | 0.886 | 3.468 | 2.071 | 10.974 |
| By tokens | 0.265 | 0.245 | 0.079 | 0.281 | 0.130 | 2.747 | 1.715 | 0.989 | 3.407 | 2.168 | 11.025 |
| $\alpha = 0.5$ | 0.236 | 0.227 | 0.129 | 0.243 | 0.165 | 2.776 | 1.732 | 0.933 | 3.444 | 2.127 | 11.012 |
| Ours | 0.246 | 0.180 | 0.124 | 0.231 | 0.218 | 2.765 | 1.762 | 0.932 | 3.443 | 2.067 | **10.969** |

Table 4: Sampling ratios and the resulting test cross-entropy loss for the 85M multilingual LM. Losses are normalized by each family's mono-family performance $\hat{\mathcal{L}}_i = \mathcal{L}_i(N, D, \boldsymbol{p})/\mathcal{L}_i^\star(N, D)$.

| | $p_{Ro}$ | $p_{Sl}$ | $p_{In}$ | $p_{Ge}$ | $p_{Si}$ | $\hat{\mathcal{L}}_{Ro}$ | $\hat{\mathcal{L}}_{Sl}$ | $\hat{\mathcal{L}}_{In}$ | $\hat{\mathcal{L}}_{Ge}$ | $\hat{\mathcal{L}}_{Si}$ | Total normalized loss |
|---|---|---|---|---|---|---|---|---|---|---|---|
| Uniform | 0.200 | 0.200 | 0.200 | 0.200 | 0.200 | 1.145 | 1.180 | 1.261 | 1.142 | 1.183 | 5.912 |
| By tokens | 0.265 | 0.245 | 0.079 | 0.281 | 0.130 | 1.123 | 1.158 | 1.407 | 1.090 | 1.238 | 6.017 |
| $\alpha = 0.5$ | 0.236 | 0.227 | 0.129 | 0.243 | 0.165 | 1.135 | 1.170 | 1.328 | 1.102 | 1.215 | 5.950 |
| Ours | 0.166 | 0.189 | 0.298 | 0.127 | 0.220 | 1.167 | 1.195 | 1.214 | 1.145 | 1.174 | **5.895** |

Lample (2019): A parameterized approach that balances between uniform and token count sampling. It introduces a parameter $\alpha$ to adjust the influence of token counts:

$$p_i = \frac{q_i^\alpha}{\sum_{j=1}^n q_j^\alpha} \text{ with } q_i = \frac{D_i}{D}.$$

Here, $\alpha = 0$ corresponds to uniform sampling, and $\alpha = 1$ corresponds to sampling by token count. Intermediate values of $\alpha$ upweigh smaller families and reduce bias towards larger ones. We use the common choice of $\alpha = 0.5$.

## 4.2 EMPIRICAL RESULTS

We demonstrate the result of using the optimal sampling ratios $\boldsymbol{p}^\star$ obtained by solving optimization problem 8. Here, we fix $D = 50B$, as we empirically find that this dataset size is sufficient to ensure near convergence across all model sizes. We first solve Eq. 8 with $N = 85M$, and verify its optimality by comparing the test cross-entropy loss of LMs trained on the resulting data mixture against the baseline methods. Next, leveraging the observed invariance of decay rates across model sizes (as shown in Section 3.4), we validate that the optimal sampling ratios derived from the 85M model generalize effectively to larger models, specifically to the 1.2B model.

**Unweighted sum.** In Table 3, we present the per-family sampling ratios, individual family losses and the total unweighted sum of loss. The multilingual LM trained on the data mixture derived from our optimal sampling ratios $\boldsymbol{p}^\star$ achieves the lowest total loss compared with other baselines. However, it is evident that language families with inherently higher magnitude of cross-entropy losses (such as Romance and Germanic) are prioritized simply because their larger magnitudes have a greater influence on the final total loss. In contrast, families with smaller inherent losses (Indic) receive less emphasis, as a reduction in their loss does not significantly impact the overall total loss.

This imbalance indicates that the unweighted sum may not be the most appropriate way to evaluate the multilingual performance, as it skews the evaluation toward families with higher inherent loss magnitudes. Consequently, while our method minimizes the unweighted total loss, it may undervalue improvements in low-loss families. To better balance performance across all language families, the following normalized loss approach offers a more fair evaluation method.

**Normalized sum.** In Table 4, we present the result for the normalized sum of losses. Unlike the unweighted sum, where the overall performance is dominated by families with higher inherent loss scales, the losses here are normalized by their mono-family performance, i.e., $\hat{\mathcal{L}}_i(N, D, p_i) = \sum_{i=1}^n \mathcal{L}_i(N, D, p_i)/\mathcal{L}_i^\star(N, D)$. This normalization ensures that each family's loss is evaluated relative to its performance ceiling, making the comparison more equitable. Note that in this setup, we expect the sampling ratios to be larger for families with larger decay rates $\gamma_i$, since the normalized loss follows the relationship $\hat{\mathcal{L}}_i(N, D, p_i) = p_i^{-\gamma_i}$. This means that prioritizing families with higher $\gamma_i$ values leads to steeper reductions in loss as sampling ratios increase, resulting in a lower total

Table 5: Sampling ratios and the resulting test cross-entropy loss for the 1.2B multilingual LM, where the total loss is computed by the unweighted sum.

| | $p_{Ro}$ | $p_{Sl}$ | $p_{In}$ | $p_{Ge}$ | $p_{Si}$ | $\mathcal{L}_{Ro}$ | $\mathcal{L}_{Sl}$ | $\mathcal{L}_{In}$ | $\mathcal{L}_{Ge}$ | $\mathcal{L}_{Si}$ | Total loss |
|---|---|---|---|---|---|---|---|---|---|---|---|
| Uniform | 0.200 | 0.200 | 0.200 | 0.200 | 0.200 | 2.349 | 1.437 | 0.728 | 3.008 | 1.714 | 9.236 |
| By tokens | 0.265 | 0.245 | 0.079 | 0.281 | 0.130 | 2.262 | 1.379 | 0.803 | 2.905 | 1.759 | 9.108 |
| $\alpha = 0.5$ | 0.236 | 0.227 | 0.129 | 0.243 | 0.165 | 2.309 | 1.414 | 0.761 | 2.963 | 1.743 | 9.190 |
| Ours (85M) | 0.246 | 0.180 | 0.124 | 0.231 | 0.218 | 2.289 | 1.424 | 0.751 | 2.953 | 1.687 | 9.104 |
| Ours (1.2B) | 0.207 | 0.182 | 0.123 | 0.249 | 0.240 | 2.313 | 1.421 | 0.752 | 2.941 | 1.674 | **9.101** |

Table 6: Sampling ratios and the resulting test cross-entropy loss for the 1.2B multilingual LM. Losses are normalized by each family's mono-family performance $\hat{\mathcal{L}}_i = \mathcal{L}_i(N, D, p_i)/\mathcal{L}_i^\star(N, D)$.

| | $p_{Ro}$ | $p_{Sl}$ | $p_{In}$ | $p_{Ge}$ | $p_{Si}$ | $\hat{\mathcal{L}}_{Ro}$ | $\hat{\mathcal{L}}_{Sl}$ | $\hat{\mathcal{L}}_{In}$ | $\hat{\mathcal{L}}_{Ge}$ | $\hat{\mathcal{L}}_{Si}$ | Total normalized loss |
|---|---|---|---|---|---|---|---|---|---|---|---|
| Uniform | 0.200 | 0.200 | 0.200 | 0.200 | 0.200 | 1.142 | 1.180 | 1.273 | 1.134 | 1.216 | 5.946 |
| By tokens | 0.265 | 0.245 | 0.079 | 0.281 | 0.130 | 1.100 | 1.133 | 1.405 | 1.095 | 1.248 | 5.982 |
| $\alpha = 0.5$ | 0.236 | 0.227 | 0.129 | 0.243 | 0.165 | 1.123 | 1.161 | 1.331 | 1.117 | 1.237 | 5.970 |
| Ours | 0.170 | 0.188 | 0.297 | 0.126 | 0.219 | 1.151 | 1.176 | 1.218 | 1.160 | 1.198 | **5.904** |

normalized loss. Indeed, the order of sampling ratios in Table 4 aligns perfectly with the order of decay rates $\gamma_i$ in Table 2. This confirms the influence of decay rates on the optimal sampling distribution.

Our optimal sampling strategy, $p^\star$ achieves the lowest total normalized loss, outperforming all other baselines. Notably, in contrast to the unweighted sum, the sampling ratios for families like Indic are significantly higher, reflecting the fact that Indic requires more data to reduce its normalized loss. Similarly, the Romance and Germanic families, which previously dominate the unweighted total loss due to their large inherent losses, now receive relatively lower sampling ratios, indicating that they need less focus in the normalized setting. This comparison demonstrates that the normalized loss provides a more balanced evaluation, ensuring that language families with smaller inherent losses are not underrepresented. Additionally, the generalization of our optimal sampling ratios across different scenarios confirms the robustness and effectiveness of our approach.

**Generalization of optimality.** We apply the optimal sampling ratios $p^\star$ derived from the 85M model to the larger 1.2B model to assess how well they generalize. From Table 5, the sampling ratios from the 85M model continue to outperform all other baselines. Furthermore, the total loss is comparable to the one achieved by the optimal sampling ratios fitted specifically for the 1.2B model. Both sets of sampling ratio exhibit a similar pattern, with lower weights for Indic (which has the least inherent loss) and higher weights for Germanic (which has the largest inherent loss).

For the normalized sum, since $\hat{\mathcal{L}}_i$ are all 1 due to normalization, the difference only lies in the decay rate $\gamma_i$. As shown in Section 3.3, $\gamma_i$ is invariant in model scales, directly leading to the same optimal sampling ratios across model sizes. From Table 6, we can see that the resulting optimal sampling ratios achieve noticeably lower total loss compared to other baselines. The results demonstrate the robustness and transferability of the optimal ratios derived from smaller models, offering a more resource-efficient strategy for large-scale multilingual LM training.

## 5 CONCLUSIONS

In this work, we develop a comprehensive and scalable scaling law for multilingual language models, addressing the complexities of training with multiple languages. Through large-scale experiments involving over 100 models across 23 languages from 5 language families, we demonstrate that the test cross-entropy loss for each language family follows a predictable power-law relationship with model size, dataset size and sampling ratios. Importantly, we introduce and validate a novel hypothesis that decouples the interaction between language families, allowing us to model the performance of each family independently of others in the training mixture. This insight enables us to derive the optimal sampling ratios that minimize the overall multilingual loss, providing a practical data mixing strategy. Our results show that the optimal sampling ratios, computed from small models, generalize effectively to models that are several orders of magnitude larger, significantly reducing the need for resource-intensive data mixture selection in large-scale training. This offers a scalable and cost-effective method for optimizing multilingual LM training and opens up new avenues for future research in multilingual scaling laws and data optimization strategies.

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

## A   DATASET DETAILS

The dataset details are introduced in Table 7. For all languages except English, we directly use the data from the CommonCrawl dataset (Conneau et al., 2020; Wenzek et al., 2020). For English data, we use a high-quality in-house dataset. For the multilingual training mixture, within each language families, we use the same smoothed sampling approach (Conneau & Lample, 2019) as introduced in Section 4.1 with $\alpha = 0.5$. Due to the massive size of the English data, we cap the proportion of English within Germanic to be 0.5. This is to ensure that the Germanic family is not dominated by English. After this capping, the "effective" total token counts for the Germanic family is around 152.48B, and we use this number for the experiments in Section 4.2. Otherwise, the proportion by token baseline will assign almost all weights to the Germanic family.

## B   MODEL SIZES AND HYPERPARAMETERS

Table 8 describes the detailed configuration of models used in our experiments.

We conduct all our experiments on NVIDIA A100 GPUs with 80GB memory. Training a 85M model with 50B tokens require approximately 256 GPU hours. According to Kaplan et al. (2020), as long as the model reaches convergence, the learning rate does not play a critical role. Thus, we do not tune the LR. For the 85M and 397M model, we start with the LR 2e-3. For the 810M and 1.2B model, we start with the LR 1e-3. Then, we apply a cosine decay schedule to decay the LR to be one-tenth of the starting LR.

## C   ADDITIONAL EXPERIMENTS

We repeat the hypothesis validation experiments as in Section 3.2 with three other language family configurations: {**Romance**, Indic, Slavic}, {**Germanic**, Romance, Slavic}, {**Sino-Tibetan**, Germanic, Slavic}, where the underlined language family has a fixed sampling ratio of either 0.2 or 0.5, and the other two families have varying ratios. From Figure 6, we still see that in all three cases, the hypothesis holds well, as the loss of the fixed family does not noticeably change when varying sampling ratios of other families. The result further validates Hypothesis 1.

Furthermore, we demonstrate the trajectory of the validation loss during training in Figure 7. This further shows that when grouping by language families, the performance of a language family only

Table 7: List of 23 languages in our study with division into 5 language families.

| Family | Per family token counts (B) | Language | Code | Per-language token counts (B) |
|---|---|---|---|---|
| Germanic | 1463.34 | English | en | 1390.50 |
| | | German | de | 52.46 |
| | | Dutch | nl | 15.25 |
| | | Danish | da | 5.13 |
| Romance | 137.43 | Spanish | es | 46.82 |
| | | French | fr | 45.98 |
| | | Italian | it | 28.93 |
| | | Romanian | ro | 12.41 |
| | | Catalan | ca | 3.29 |
| Slavic | 126.77 | Russian | ru | 95.29 |
| | | Ukrainian | uk | 20.55 |
| | | Slovak | sk | 5.57 |
| | | Serbian | sr | 2.87 |
| | | Croatian | hr | 2.49 |
| Indic | 40.86 | Hindi | hi | 12.76 |
| | | Bengali | bn | 9.49 |
| | | Nepali | ne | 2.20 |
| | | Marathi | mr | 2.02 |
| | | Tamil | ta | 5.56 |
| | | Telugu | te | 2.82 |
| | | Kannada | kn | 2.37 |
| | | Malayalam | ml | 3.65 |
| Sino-Tibetan | 67.41 | Chinese | zh | 67.41 |

Table 8: Model configurations.

| # Layers | Hidden size | # Heads | Head dim | Total parameters |
|---|---|---|---|---|
| 12 | 768 | 6 | 128 | 85,056,768 |
| 14 | 1536 | 12 | 128 | 396,645,248 |
| 21 | 1792 | 14 | 128 | 809,732,672 |
| 24 | 2048 | 16 | 128 | 1,208,604,160 |

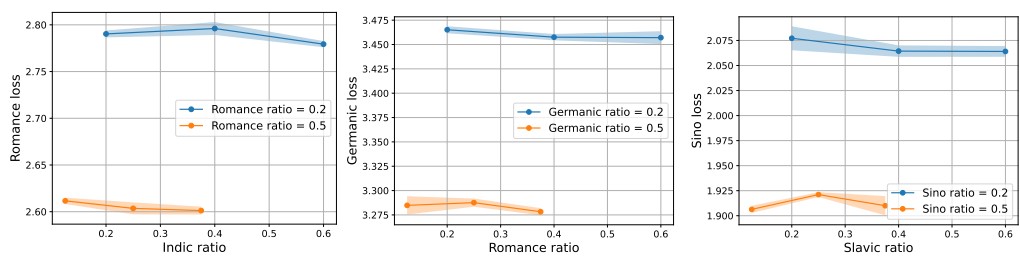

Figure 6: Further validation on three family combinations: {**Romance**, Indic, Slavic} (left), {**Germanic**, Romance, Slavic} (middle), {**Sino-Tibetan**, Germanic, Slavic} (right). In all three cases, the loss for the family with a fixed ratio does not vary with proportions of other jointly trained family.

depends on its own sampling ratio, regardless of the combinations of other jointly trained families. In contrast, cross-group transfer is evident when the languages within one family is split into two groups.

## D FULL FITTING RESULTS

### D.1 FITTING $\mathcal{L}_i(p_i)$

We present the full power law fitting for $\mathcal{L}_i(p_i)$ across all model sizes in Figures 8 to 11. For all model sizes, the fitting quality is consistent with low R-squared. For each language, the exponent is essentially invariant, further validating our claim. All fittings are performed with an off-the-shelf function fitting tool (`scipy.optimize.least_squares`).

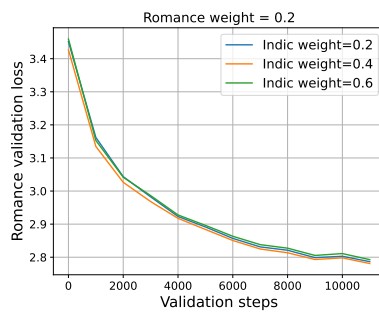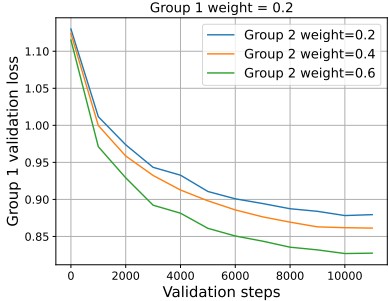

Figure 7: **Left:** The trajectories of the Romance validation loss are almost identical for different weight combinations of other jointly trained families. **Right:** In contrast, when both groups contain Indic languages, Group 1 loss is clearly higher when the Group 2 sampling ratio is lower, indicating significant cross-group transfer.

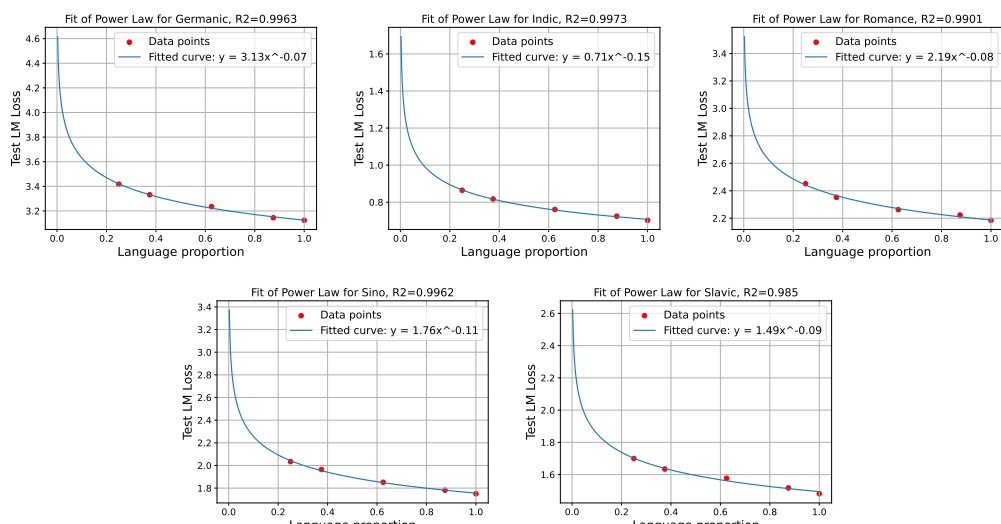

Figure 8: Power law fitting for 5 language families on the 85M model.

## D.2 FITTING $\mathcal{L}_i(N, D, p_i)$

Table 9: Values of the fitted coefficients for all families. The validation value on the last column $\mathcal{L}_i^*$ is calculated using $N = 397M, D = 50B$. Also values assume $N$ in Millions and $D$ in Billions.

| Language family | $E$ | $A$ | $B$ | $\alpha$ | $\beta$ | $\gamma$ | val $\mathcal{L}_i^*$ |
|---|---|---|---|---|---|---|---|
| Romance | 1.303 | 2.509 | 2.186 | 0.229 | 0.557 | 0.078 | 2.186 |
| Slavic | 0.001 | 1.561 | 1.240 | 0.186 | 0.112 | 0.093 | 1.311 |
| Indic | 0.001 | 0.782 | 0.691 | 0.194 | 0.152 | 0.140 | 0.626 |
| Germanic | 1.696 | 2.708 | 2.045 | 0.192 | 0.512 | 0.065 | 2.829 |
| Sino-Tibetan | 0.243 | 2.018 | 1.010 | 0.143 | 0.211 | 0.115 | 1.542 |

We present the full list of our fitted parameters in Table 9. The fitting is done by jointly estimating $E, A, B, \alpha, \beta, \gamma$ together. Thus, the values slightly deviate from Table 2. One can alternatively fit $\gamma$ first, as it is the only parameter that requires empirical data with varying sampling ratios $p$. Then, for all of $E, A, B, \alpha, \beta$, they can be fitted purely with mono-family losses.

We produce similar plots as Figure 5 for all language families in Figure 12. Overall, the scaling law captures the relationship very well.

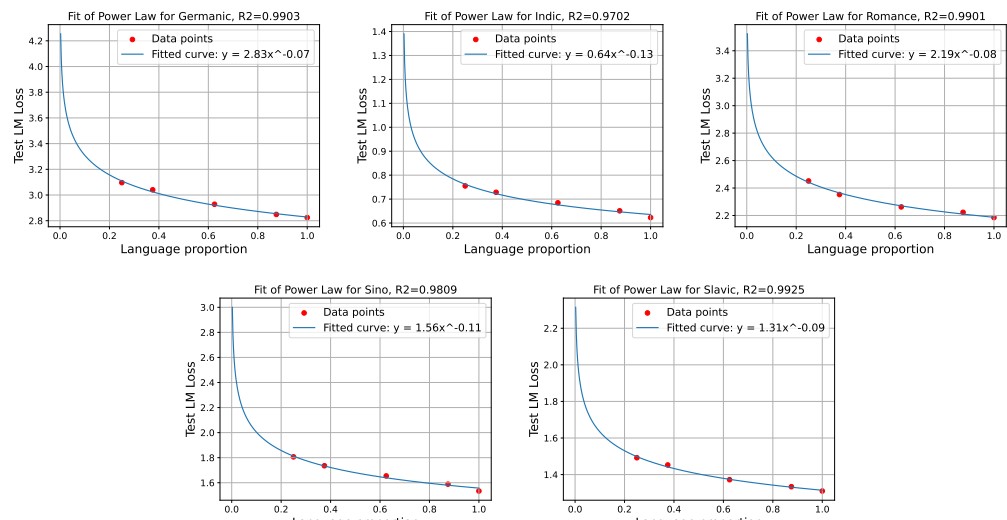

Figure 9: Power law fitting for 5 language families on the 397M model.

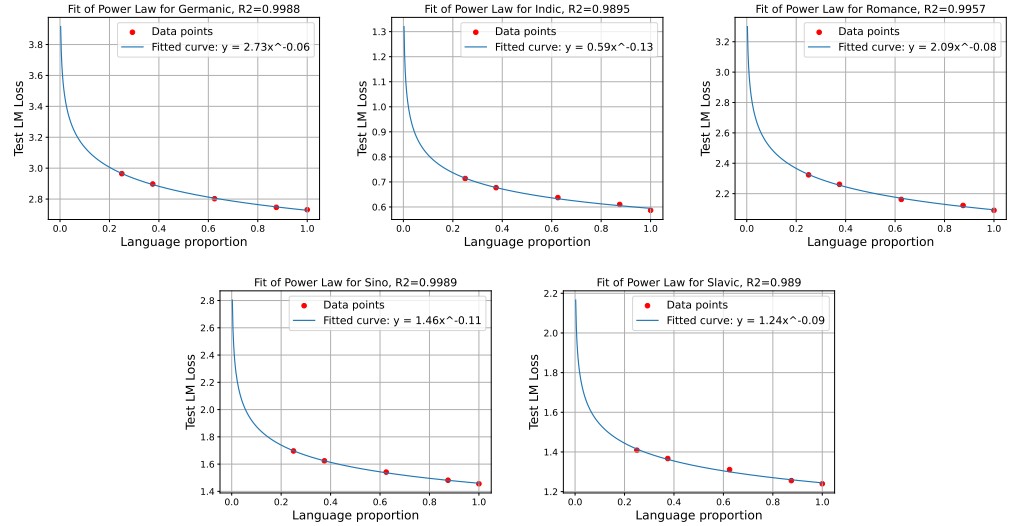

Figure 10: Power law fitting for 5 language families on the 810M model.

# E   JUSTIFICATION ON THE FORM OF $\mathcal{L}_i(N, D, p)$

From Section 3.4, we already know that the relationship between $\mathcal{L}, N, D, \boldsymbol{p}$ follow the form

$$\mathcal{L}_i(N, D, p_i) = \mathcal{L}_i^\star(N, D) \cdot p_i^{-\gamma_i}. \tag{9}$$

For simplicity, we omit the subscript $i$ for $E_i, A_i, B_i, \alpha_i, \beta_i, \gamma_i, \mathcal{L}_i$, which are family dependent.

Recall the Chinchilla scaling law

$$\mathcal{L}(N, D) = E + \frac{A}{N^\alpha} + \frac{B}{D^\beta} \tag{10}$$

Since we know that separate scaling laws are valid for given $p$, in the general form, the parameters in Eq. 10 can be dependent on the sampling ratio $p$:

$$\mathcal{L}(N, D, p) = E(p) + \frac{A(p)}{N^{\alpha(p)}} + \frac{B(p)}{D^{\beta(p)}} \tag{11}$$

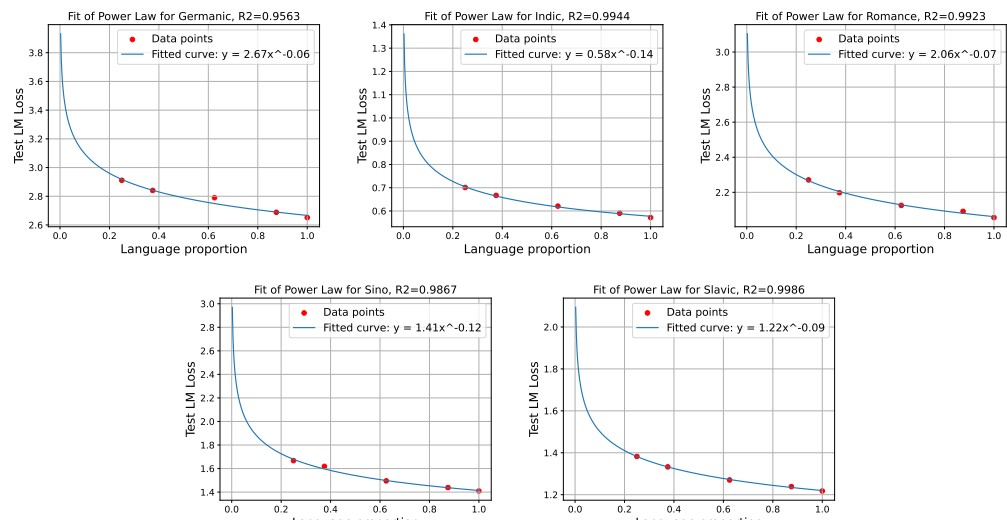

Figure 11: Power law fitting for 5 language families on the 1.2B model.

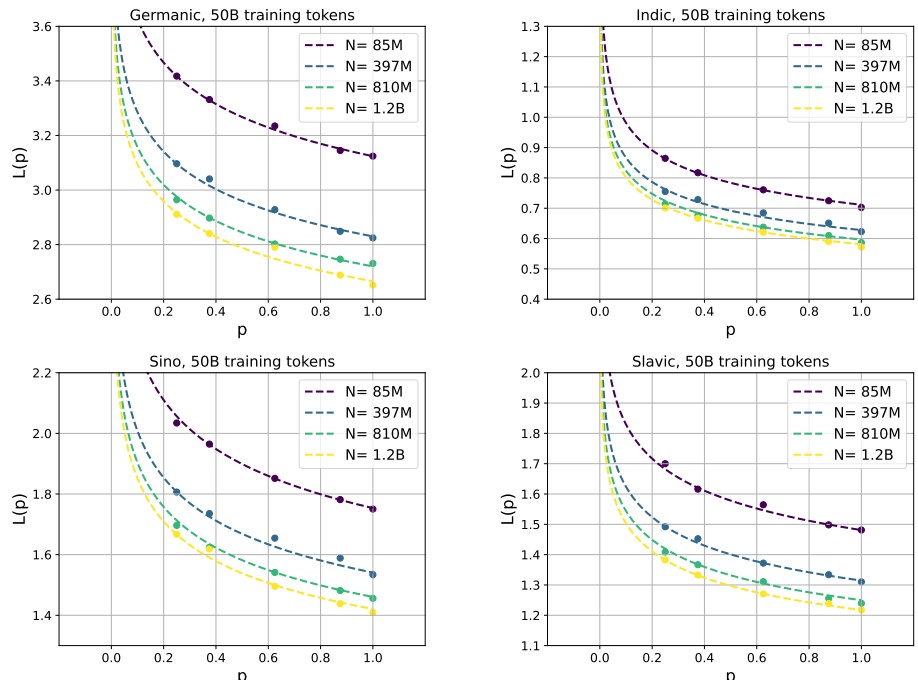

Figure 12: Joint power law fitting for 4 language families at 50B tokens across all model sizes.

Following the above observation that models with given $p$ obey Chinchilla scaling laws given by Eq. 10, the key question that arises is how the general notion of sampling ratio $p$ can be incorporated into the joint scaling law. Moreover, the scaling law formula from Eq. 11 for constant $N$ and $D$ has to be representable by Eq. 9. It is anticipated to align with the latter, consisting of distinct power laws, each with specific parameters for different $N$ and $D$ values. Consequently, the objective is to identify a function that fulfills these criteria

$$\mathcal{L}(N, D, p) = \mathcal{L}^{\star}(N, D)p^{-\gamma(N,D)} = E(p) + \frac{A(p)}{N^{\alpha(p)}} + \frac{B(p)}{D^{\beta(p)}} \tag{12}$$

With this in mind, we aim to determine which of these parameters (on RHS) remain independent of $p$ and identify the functional form of the others.

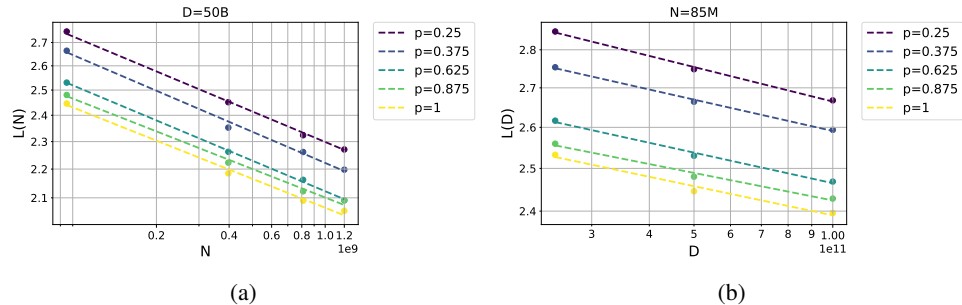

(a)                                                     (b)

Figure 13: **(a)** For a fixed token count, there is a linear relationship between $\log \mathcal{L}$ and $\log(N)$ for different values of sampling ratio $p$. Both axes are in the log-scale. **(b)** For a fixed model size, there is a linear relationship between $\log \mathcal{L}$ and $\log(D)$ for different values of sampling ratio $p$. Both axes are in the log-scale. The lines are nearly parallel, indicating that $\alpha$ and $\beta$ do not depend on $p$.

**Model Size $N$ for different Sampling Ratio $p$.** As seen in Figure 13a, we observe the linear relationship between $\log N$ and $\log \mathcal{L}$ and since the lines are parallel for any given $p$, the slope $\alpha$ (Eq. 11) is independent of the sampling ratio $p$. Therefore we can assume that $\alpha(p) = \alpha$ is constant. [6]

**Dataset size $D$ for different Sampling Ratio $p$.** As seen in Figure 13b, we observe the linear relationship between $\log D$ and $\log \mathcal{L}$ and since the lines are parallel for any given $p$, the slope $\beta$ (Eq. 11) is independent of the sampling ratio $p$. Therefore we can assume that $\beta(p) = \beta$ is constant.

**Infinite model size $N$ and dataset size $D$.** Consider the limit of Eq. 12 where $N, D \to \infty$, we get the functional form of $E(p)$:

$$E(p) = \mathcal{L}^{\star}(\infty, \infty)p^{-\gamma} = c_E p^{-\gamma} \tag{13}$$

where $c_E := \mathcal{L}^{\star}(\infty, \infty)$ is a constant.

**Infinite dataset size $D$.** Consider the limit of Eq. 12 where $D \to \infty$, plugging in Eq. 13, taking logs of both sides and moving sides:

$$\log A(p) + \gamma \log p = \alpha \log N + \log(c_{A'}(N) - c_E) \tag{14}$$

where $c_{A'}(N) := \mathcal{L}^{\star}(N, \infty)$ only depends on $N$.

The LHS of Eq. 14 only depend on $p$, whereas the RHS only depends on $N$ so they should both equal some constant,$c_A$ (this step relies on our proof above that $\alpha, \beta$ and $\gamma$ are independent of $N, D$ and $p$), resulting in the functional form of $A(p)$

$$A(p) = c_A p^{-\gamma} \tag{15}$$

**Infinite dataset size $N$.** We can consider the limit of Eq. 12 where $N \to \infty$ and by a symmetric argument to above ($D$ instead of $N$) we get the functional form of $B(p)$

$$B(p) = c_B p^{-\gamma} \tag{16}$$

Plugging the functional forms of $E(p), A(p)$ and $B(p)$ and reverting back to $E, A, B$ instead of $c_E, c_A, c_B$ and adding back the omitted subscript $i$, we obtain the final functional form for the joint scaling law:

$$\mathcal{L}_i(N, D, p_i) = \left( E_i + \frac{A_i}{N^{\alpha_i}} + \frac{B_i}{D^{\beta_i}} \right) p_i^{-\gamma_i}.$$

---

[6]Strictly speaking, for a fixed $D$, $\log\left(\mathcal{L}(N) - c_{B'}(p)\right) = \log A(p) - \alpha(p) \log N$ where $c_{B'}(p)$ is a constant dependant on $p$. However, note that $\log\left(\mathcal{L}(N)\right) - \frac{c_{B'}(p)}{\mathcal{L}(N)} \leq \log\left(\mathcal{L}(N) - c_{B'}(p)\right) \leq \log\left(\mathcal{L}(N)\right)$. Hence $-\alpha(p) \log(N) + \log A(p) \leq \log\left(\mathcal{L}(N)\right) \leq -\alpha(p) \log(N) + \log A(p) + \frac{c_{B'}(p)}{\mathcal{L}(N)}$. Empirically, we found $\log\left(\mathcal{L}(N)\right) = m \log(N) + c$ to fit well, i.e for large enough N, $\frac{c_{B'}(p)}{\mathcal{L}(N)}$ behaves more or less like a constant. Because of that, we assume the functional form of $(\log\left(\mathcal{L}(N)\right) = -\alpha(p) \log(N) + c(p))$ ansatz. The fact that $\alpha(p)$ is independent of p follows from the fact that for different values of p, the slope is constant.

## F  OPTIMAL SAMPLING RATIOS ANALYTIC SOLUTION

We use Lagrange multipliers to find the point of minimum weighted loss, $\boldsymbol{p}^\star$. We set $\Lambda(\boldsymbol{p}, \lambda) = \mathcal{L} + \lambda g$ and require that

$$\frac{\partial \Lambda(\boldsymbol{p}, \lambda)}{\partial \boldsymbol{p}} = 0$$
$$\frac{\partial \Lambda(\boldsymbol{p}, \lambda)}{\partial \lambda} = 0$$

(17)

which gives a system of $n$ equations $i = 1, \ldots, n$ such that:

$$\frac{\partial}{\partial p_i} \left[ \sum_i w_i \mathcal{L}_i^\star p_i^{-\gamma_i} + \lambda \left( \sum_i p_i - 1 \right) \right] = 0,$$

(18)

Carrying out the differentiation, together with the constraint results in a system of $n + 1$ equations with $n + 1$ variables $(p_1, \ldots, p_n, \lambda)$ that we can solve:

$$-w_i \mathcal{L}_i^\star \gamma_i p_i^{-(1+\gamma_i)} + \lambda = 0, \text{ for } i = 1, \ldots, n$$
$$p_1 + \cdots + p_n = 1$$

(19)

Rearranging the first $n$ equations of Eq. 19 we get:

$$p_i = (w_i \mathcal{L}_i^\star \gamma_i)^{\frac{1}{1+\gamma_i}} \lambda^{-\frac{1}{1+\gamma_i}}$$

(20)

and plugging it to the last equation:

$$\sum_{i=1}^n (w_i \mathcal{L}_i^\star \gamma_i)^{\frac{1}{1+\gamma_i}} \lambda^{-\frac{1}{1+\gamma_i}} - 1 = 0$$

(21)

And we are done with proving the implicit solution. It is worth noting that the polynomial in Eq. 21 is called an exponential polynomial Ritt (1929), however considering the solution to those is out of the scope of this paper.

The bordered Hessian matrix in this case is:

$$\mathbf{H} = \begin{bmatrix} 0 & \frac{\partial g}{\partial p_1} & \frac{\partial g}{\partial p_2} & \cdots & \frac{\partial g}{\partial p_n} \\ \frac{\partial g}{\partial p_1} & \frac{\partial^2 \Lambda}{\partial p_1^2} & \frac{\partial^2 \Lambda}{\partial p_1 \partial p_2} & \cdots & \frac{\partial^2 \Lambda}{\partial p_1 \partial p_n} \\ \frac{\partial g}{\partial p_2} & \frac{\partial^2 \Lambda}{\partial p_2 \partial p_1} & \frac{\partial^2 \Lambda}{\partial p_2^2} & \cdots & \frac{\partial^2 \Lambda}{\partial p_2 \partial p_n} \\ \vdots & \vdots & \vdots & \ddots & \vdots \\ \frac{\partial g}{\partial p_n} & \frac{\partial^2 \Lambda}{\partial p_n \partial p_1} & \frac{\partial^2 \Lambda}{\partial p_n \partial p_2} & \cdots & \frac{\partial^2 \Lambda}{\partial p_n^2} \end{bmatrix} = \begin{bmatrix} 0 & 1 & 1 & \cdots & 1 \\ 1 & \frac{\partial^2 \Lambda}{\partial p_1^2} & 0 & \cdots & 0 \\ 1 & 0 & \frac{\partial^2 \Lambda}{\partial p_2^2} & \cdots & 0 \\ \vdots & \vdots & \vdots & \ddots & \vdots \\ 1 & 0 & 0 & \cdots & \frac{\partial^2 \Lambda}{\partial p_n^2} \end{bmatrix}$$

(22)

with

$$\frac{\partial^2 \Lambda}{\partial p_i^2} = w_i \mathcal{L}_i^\star \gamma_i (1 + \gamma_i) p_i^{-(2+\gamma_i)}$$

(23)

we have

$$|\mathbf{H}| = - \sum_{i=1,\ldots,n} \frac{\partial^2 \Lambda}{\partial p_1^2} \frac{\partial^2 \Lambda}{\partial p_2^2} \cdots \frac{\partial^2 \Lambda}{\partial p_{i-1}^2} \frac{\partial^2 \Lambda}{\partial p_{i+1}^2} \cdots \frac{\partial^2 \Lambda}{\partial p_n^2}$$

(24)

Each of the terms on 23 is positive for all $p_i \in (0, 1]$ ($\mathcal{L}_i^\star, \gamma_i > 0$) so the determinant of the bordered Hessian is a negative sum of positive products which is always negative meaning the solution we find on equation 19 is a always minimum as required.

Now consider the first order Taylor polynomial for $\lambda^{-\frac{1}{1+\gamma_i}}$ around $\gamma_i = 0$

$$\left(\frac{1}{\lambda}\right)^{\frac{1}{1+\gamma_i}} = \frac{1}{\lambda} + \gamma_i \frac{\log \lambda}{\lambda} + \mathcal{O}(\gamma_i^2)$$
$$\approx \frac{1}{\lambda} + \gamma_i \frac{\log \lambda}{\lambda} \tag{25}$$

Plugging this back into Eq. 21, we get

$$\sum_{i=1}^{N} (w_i \mathcal{L}_i^\star \gamma_i)^{\frac{1}{1+\gamma_i}} \left(\frac{1}{\lambda} + \gamma_i \frac{\log \lambda}{\lambda}\right) = 1$$
$$\sum_{i=1}^{N} (w_i \mathcal{L}_i^\star \gamma_i)^{\frac{1}{1+\gamma_i}} + \log \lambda \left(\sum_{i=1}^{N} \gamma_i (w_i \mathcal{L}_i^\star \gamma_i)^{\frac{1}{1+\gamma_i}}\right) = \lambda \tag{26}$$

Finally, taking the first order Taylor polynomial for $\log \lambda$ around $\lambda = 1$, we have

$$\log \lambda = (\lambda - 1) + \mathcal{O}((\lambda - 1)^2)$$
$$\approx (\lambda - 1) \tag{27}$$

Plugging this back into Eq. 26, we get

$$\sum_{i=1}^{N} (w_i \mathcal{L}_i^\star \gamma_i)^{\frac{1}{1+\gamma_i}} + (\lambda - 1) \left(\sum_{i=1}^{N} \gamma_i (w_i \mathcal{L}_i^\star \gamma_i)^{\frac{1}{1+\gamma_i}}\right) = \lambda$$
$$\lambda = \frac{\sum_{i=1}^{N} (w_i \mathcal{L}_i^\star \gamma_i)^{\frac{1}{1+\gamma_i}} (1 - \gamma_i)}{1 + \sum_{i=1}^{N} \gamma_i (w_i \mathcal{L}_i^\star \gamma_i)^{\frac{1}{1+\gamma_i}}} \tag{28}$$
$$p_i = \left(\frac{w_i \mathcal{L}_i^\star \gamma_i \left(1 + \sum_{i=1}^{N} \gamma_i (w_i \mathcal{L}_i^\star \gamma_i)^{\frac{1}{1+\gamma_i}}\right)}{\sum_{i=1}^{N} (w_i \mathcal{L}_i^\star \gamma_i)^{\frac{1}{1+\gamma_i}} (1 - \gamma_i)}\right)^{\frac{1}{1+\gamma_i}}$$

For small $\gamma_i$ that also agrees with the zero order approximation

$$p_i \approx \frac{w_i \mathcal{L}_i^\star \gamma_i}{\sum_{i=1}^{n} w_i \mathcal{L}_i^\star \gamma_i}. \tag{29}$$

Table 10: Comparison of optimal sampling ratios obtained from the numerical solver and the analytical solution.

|  | $p_{Ro}$ | $p_{Sl}$ | $p_{In}$ | $p_{Ge}$ | $p_{Si}$ |
|---|---|---|---|---|---|
| Numerical solver (85M) | 0.246 | 0.180 | 0.124 | 0.231 | 0.218 |
| Analytical solution (85M) | 0.243 | 0.168 | 0.121 | 0.240 | 0.226 |
| Numerical solver (1.2B) | 0.207 | 0.182 | 0.123 | 0.249 | 0.240 |
| Analytical solution (1.2B) | 0.205 | 0.161 | 0.122 | 0.253 | 0.259 |

Here, we demonstrate that this approximate analytical solution has similar resulting optimal sampling ratios as the numerical solution. As in Table 10, the sampling ratios are indeed similar.

## G LIMITATIONS

One limitation of our work is that while the scaling law predicts the performance well in most cases, its utility may diminish when dealing with very small $N$, $D$ and $\boldsymbol{p}$ values. For instance, using the formulation in Eq. 2, setting $p_i = 0$ results in an infinite loss. However, in reality, even if a training mixture contains no data for a particular family, its test cross-entropy loss is likely to be non-vacuous, as some general language information can still be transferable across families. In such cases, the resulting loss will be entirely dependent on cross-family transfer, which is challenging to quantify. For

example, intuitively, if the test language family is Germanic, the Romance mono-family model might perform noticeably better than the Sino-Tibetan mono-family model, due to linguistic similarities. Consequently, there may not be a fixed value associated with $\mathcal{L}_i(p_i = 0)$ in our formulation.

Similarly, for small $D$ values, the risk of overfitting may need to be explicitly modeled, as suggested in Chen et al. (2024). Note that this issue is more pronounced when modeling individual languages, as low-resource languages suffer more from limited data. In contrast, language families are less affected due to the abundance of data within each family.

On the other hand, small $N$ values are less problematic, as scaling laws are primarily intended to predict performance for larger models. We have demonstrated that our formulation works well even for relatively small models, such as those with 85M parameters.

