# OpenReview forum: "Scaling Laws for Multilingual Language Models"
_ICLR.cc/2025/Conference — Submitted to ICLR 2025_

### Official Review · Reviewer_Czqd · 2024-10-25

**Soundness:** 1
**Presentation:** 3
**Contribution:** 2
**Rating:** 5
**Confidence:** 3

**Summary:**

This paper presents a scaling law for multilingual LLMs. It also argues for a hypothesis that the cross entropy loss for each language family in the LLM is determined only by the sampling ratio of the family. Experiments are performed on 23 languages (from 5 families) and model sizes range from 95M parameters to 1.2B parameters.

**Strengths:**

1. Scaling laws are important objects of study; better understanding can help us inform modeling/training decisions. The paper examines scaling law in the context of multilingual LLM, which I believe is new. There are scaling laws on monolingual LLM and machine translation models, but this setup deserves its own study.

2. One practical result is the prediction of good language sampling ratios. This is an important design decision.

**Weaknesses:**

1. While the results are interesting, I believe the claims are too strong. It would be better to tone down. Here are some examples of strong claims:
- page 1: "Our contributions are grounded ... making our analysis scalable to an arbitrary number of languages." - The experiments are performed on 23 languages. There are 6000+ languages in the world.
- page 7: Half a page is dedicated to explaining how the work is better than previous work (i.e. generalized framework, multilingual scalability, incorporation of dataset size, simpler...) I think it suffices to keep it simple and say that your work looks at a different problem; I do not see a need to argue so much. Further, it is somewhat strange to argue that your equation is simpler than previous work -- if we view scaling law as a phenomenon to be discovered, then faithfulness is probably more important than simplicity perhaps?

2. I question whether the proposed argument about language families as the main factor. It is true that there is probably more crosslingual transfer within a language family and less across families, but this is a very simplified view of language. Specifically:
- Language families are genetic relationships, but in practice there are significant areal effects (c.f. https://aclanthology.org/N09-1067/ ). There are also other kinds of sharing, for example Japanese is an isolate but there are many Kanji characters borrowed from Chinese.
- The experiments are actually not very diverse in terms of language families: Of the 23 languages (5 families) used, 22 languages (4 "families") are actually from the Indo-European "family". Germanic, Romance, Slavic, Indic are considered different branches of the Indo-European "family". So only 1 language (Chinese) is truly from a different family in the linguistic sense.
- One complication is script, which is not discussed in the paper. Different scripts will lead to very different tokenization results, which likely impacts LLM training. In the languages considered, most use the Latin script and the Russian branch uses both Cyrillic and Latin, I assume.

**Questions:**

1. This is not necessary a weakness, but I wonder if we should view the models to be on the small side compared to contemporaneous work. For example, training data is up to 50B tokens and model size is 398M parameters to 1.2B parameters. I think there is value in the experiments in any case, but can you comment a bit on the scale of some of the large multilingual LLMs in the literature? For example, EuroLLM-1.7B is trained on 4 trillion tokens.

2. The citations are generally adequate. Here are a two more suggested papers to add - they discuss scaling laws for bilingual machine translation.
https://aclanthology.org/2021.emnlp-main.478/
https://arxiv.org/abs/2109.07740

3. Regarding sampling strategy, the general rule-of-thumb I think is to up-weight lower-resourced languages. At the same time, some anecdotes seem to suggest it is better to make English data the largest (e.g. 50%) due to its potential to help LLM in reasoning and other tasks. Can you comment about the chosen sampling strategy in your experiments in the context of these guidelines? How similar or different is it? Or does it depend on the group of languages?

---

> ### Author Response · Authors · 2024-11-20
>
> **[Strong claims – page 1]** Our intent is to convey that the proposed framework is designed to be applicable to a large number of languages, as long as they meet our minimal cross-group transfer and data sufficiency requirements. While we indeed test our approach on 23 languages, the framework itself is general enough to adapt to additional languages under similar conditions. We hope this explanation clarifies this distinction and better reflect the potential, rather than exhaustive, scalability of our approach.
>
> ---
> **[Strong claims – page 7]** Our aim with this section is to provide context for common settings and challenges in multilingual scaling, and to clarify how our approach addresses these by highlighting differences in functional forms and applications. We include this comparison as a substitute for a traditional “related works” section, as the work we compare against is one of the few representatives in this field and shares similar use cases and problem formulations with related studies. We hold the prior work in high regard, as it provides an essential foundation for our contributions.
>
> In addition, we use the term “simple and predictive” in the paper, indicating that the simplification not only does not sacrifice faithfulness, but actually enhances utility. This is demonstrated by the accuracy of scaling law fitting as well as the optimized sampling ratios.  In line with Occam’s Razor, we find that the simpler explanation proves to be more effective in this context.
>
> ---
> **[Alternative grouping strategies]** Please refer to the second point in our general response. In the first point of our general response, we also provide additional evidence about the validity of language-family-based grouping.
>
> ---
> **[Language diversity]** Our language selection follows the practice in [1] for representativeness and easier evaluation. Furthermore, upon revisiting the composition of the CC-100 dataset, we find that the only other major language family not covered in our experiments is the Afro-Asiatic family (e.g., Arabic). Unfortunately, due to current limitations in available multilingual datasets, expanding our experiments to the diversity level the reviewer suggests is challenging. We hope future efforts will continue to enhance the multilingual dataset landscape, as more diverse language coverage would allow even broader validation of our proposed scaling law. Based on the consistent results observed across our chosen language families, we are optimistic that our findings will hold on a more diverse language corpus.
>
> [1] Okapi: Instruction-tuned Large Language Models in Multiple Languages with Reinforcement Learning from Human Feedback. Lai et al. 2023.
>
> ---
> **[Tokenization]** While we acknowledge that tokenizer vocabulary imbalance is a key issue for multilingual LMs in general, this is not a problem specific to our setting. We actually discuss this problem in our paper (line 423, page 8), which motivates our use of the normalized losses for evaluation. In general, tokenization for high-resource languages tends to align more with semantic units, whereas low-resource languages often experience less meaningful tokenization due to data imbalance during tokenizer training. This can indeed lead to higher LM losses for languages with larger tokenizer vocabulary. Nevertheless, improvement to multilingual tokenizers is not the focus of our work, so we will leave it for future works in related areas.
>
> ---
> **[Larger models]** We acknowledge that the experiment scale is on the smaller side compared with SoTA LLMs, which usually starts at 7B parameters. However, our model sizes are still comparable to or even larger than those used in recent studies of scaling laws, such as [1] (40M to 1.5B parameters), [2] (20M to 1B parameters), and [4] (64M to 312M parameters).
>
> When considering data scales of EuroLM (4T tokens), practical limitations arise due to model learning capacities. For instance, training an 85M model on 4T tokens would likely be an inefficient allocation of compute. Our experiments, therefore, focus on data and model size combinations that are computationally practical and reflective of the goal of scaling laws: to extrapolate performance predictions at larger scales by **training on smaller models**. In addition, using more tokens could push us to a data-constrained regime, as low-resource languages might be trained for more epochs to adjust to the larger token horizon, possibly leading to overfitting. This regime often requires specific treatment [3, 4], which is beyond the scope of our paper.
>
> [1] Scaling Laws for Neural Machine Translation. Ghorbani et al. 2021.
>
> [2] Scaling Laws for Multilingual Neural Machine Translation. Fernandes et al. 2023.
>
> [3] Scaling Data-Constrained Language Models. Muennighoff et al. 2023.
>
> [4] On the Pareto Front of Multilingual Neural Machine Translation. Chen et al. 2023.

---

> > ### Author Response · Authors · 2024-11-20
> >
> > **[Sampling strategies]** Our study of the sampling ratios is based on the training objective. While up-weighting lower-resourced languages may indeed benefit use cases where all languages are considered equally important, other strategies may better serve scenarios where particular languages (e.g., English) are prioritized for their broader utility. Our approach allows for flexibility by incorporating importance weights ($w$ in Eq. (1)), enabling users to emphasize specific language families as needed. This adaptability is evident in our results: the sampling ratios vary significantly across tables, such as between Table 3 and Table 4, and between Table 5 and Table 6. These differences reflect how our method can adjust sampling strategies based on the importance assigned to each language group, aligning with different practical needs and priorities.

---

> ### Author Response · Authors · 2024-11-25
>
> Dear reviewer Czqd,
>
> We sincerely appreciate the valuable insights you have provided in the review, and we have diligently worked to address your comments through our rebuttal. As a kind reminder, **the author-review discussion period will end tomorrow**. Should you have any further inquiries or seek additional clarifications, please do not hesitate to let us know. If the questions have been resolved, we would greatly appreciate it if you could acknowledge our response and consider updating the score.
>
> Thank you again for your time and effort!
>
> Authors

---

> ### Author Response · Authors · 2024-12-02
>
> Dear reviewer  Czqd,
>
> We sincerely appreciate the valuable insights you have provided in the review, and we have diligently worked to address your comments through our rebuttal. As a kind reminder, **tomorrow is the last day that the reviewer can post messages to authors**. Please do not hesitate to let us know if you have further inquiries. If the questions have been resolved, we would greatly appreciate it if you could acknowledge our response and consider updating the score.
>
> Thank you again for your time and effort!
>
> Authors

---

### Official Review · Reviewer_ig2E · 2024-10-28

**Soundness:** 2
**Presentation:** 3
**Contribution:** 2
**Rating:** 5
**Confidence:** 3

**Summary:**

The paper studies a scaling law for general-purpose decoder-only multilingual Language Models to reduce the need for resource-intensive data mixture selection in large-scale training. This paper focuses on balancing language families rather than individual languages during pretraining. The authors hypothesize that pretraining can leverage language family groupings to simplify the complexity of multilingual learning. From this hypothesis, they derive a power-law relationship that links dataset size, model size, and language sampling ratios, helping to optimize training across multiple scales (85M, 397M, 810M, and 1.2B parameters) on each of the 5 language families.

**Strengths:**

S1. The proposed scaling law could simplify the complexity of cross-lingual interactions during training, offering a meaningful extension of existing work that focuses primarily on monolingual or bilingual models. This decoupling from individual languages could also reduce the computational burden.

S2. The paper also proposed an optimal sampling ratio as a practical data-mixing strategy, which could be applicable for real-world multilingual model training.

S3. The paper is clear in its presentation, with a well-structured flow of ideas that makes it easy to follow. The authors provide clear and well-presented tables, figures, and mathematical formulas, which support the text and make complex concepts more digestible.

**Weaknesses:**

W1. The primary issue with the paper is its focus on the generality of the proposed scaling law without applying it to specific downstream tasks (beyond machine translation as it is claimed). Testing it on downstream tasks like summarization or question answering, on metrics that are widely used to assess the performance of each downstream task, would have strengthened the claim that this scaling law is for a general-purpose decoder multilingual language model. It also provides more solid evidence of its applicability across different use cases to interested readers.

W2. The improvement in normalized loss from the optimal sampling ratio appears minimal. For example, in Table 4 for the 85M model, the difference between the uniform sampling ratio ($5.912$) and the proposed sampling ratio ($5.895$) is just $0.017$, raising questions about the significance of this gain. It is unclear to me how beneficial such a small reduction in loss is, especially given the additional complexity introduced by the data-mixing strategy. Again, going back to W1, if these minor improvements do not translate into meaningful performance gains on downstream tasks, the value of this approach is not that convincing. The paper should showcase the practical advantages of achieving these marginal gains in normalized loss.

Additionally, in Table 6 for the 1.2B model, the total normalized loss for uniform sampling seems inconsistent: summing the individual values ($1.142 + 1.180 + 1.273 + 1.142 + 1.134$) results in $5.871$, which differs from the reported value of $5.946$. If the former is true, this could imply that the proposed sampling ratio doesn't generalize as well for larger models (1.2B), which is the main purpose of this proposed sampling ratio. Clarifying this would help substantiate whether the benefits of the proposed ratio are meaningful across different model sizes.

W3. While the paper provides some empirical evidence supporting Hypothesis 1, this evidence is limited to specific language families (Indic-Romance and Slavic-Sino-Tibetan, with Sino-Tibetan only represented by Chinese). This is a substantial claim and is referenced throughout the paper. However, there are numerous studies, particularly in monolingual settings, showing cross-lingual transfer outside of phylogenetically related language families [1]. For example, English (Germanic) or other high-resource languages are widely used for cross-lingual transfer in various multilingual setups [1, 2]. Language similarity can also be measured in multiple ways, such as typological or geographical distance, and not solely by language families [3]. It would be valuable to investigate whether the proposed hypothesis holds between languages like English and Romance languages, where there is known cross-lingual transfer despite family differences.

### References:

[1] Lin, Y.-H., Chen, C.-Y., Lee, J., Li, Z., Zhang, Y., Xia, M., Rijhwani, S., He, J., Zhang, Z., Ma, X., et al. (2019). Choosing transfer languages for cross-lingual learning. *arXiv preprint arXiv:1905.12688*. [Link to Paper](https://arxiv.org/abs/1905.12688)

[2] Adelani, D. I., Liu, H., Shen, X., Vassilyev, N., Alabi, J. O., Mao, Y., Gao, H., Lee, A. E.-S. (2023). SIB-200: A simple, inclusive, and big evaluation dataset for topic classification in 200+ languages and dialects. *arXiv preprint arXiv:2309.07445*. [Link to Paper](https://arxiv.org/abs/2309.07445)

[3] Littell, P., Mortensen, D. R., Lin, K., Kairis, K., Turner, C., Levin, L. (2017). URIEL and lang2vec: Representing languages as typological, geographical, and phylogenetic vectors. *Proceedings of the 15th Conference of the European Chapter of the Association for Computational Linguistics: Volume 2, Short Papers*, 8–14. [Link to Paper](https://aclanthology.org/E17-2002/)

**Questions:**

In addition to the points raised in the weaknesses, I do have additional questions:

- Could you provide more details by comparing your proposed scaling law with previous work in multilingual models? For example, can you fit previous works on scaling laws into your framework and compare their performance? This would give readers better clarity on the benefits and advantages of using the proposed approach.
- Have you tested your scaling law on out-of-distribution data as in data outside the CommonCrawl dataset? Since your test set is taken from the same dataset, I wonder if the scaling law would generalize well to datasets that are not part of CommonCrawl, especially as you included dataset size as part of the proposed scaling law. Exploring the model’s generalization ability on such data could provide additional insight.
- How well does the scaling law apply to low-resource language families like Niger-Congo? Also, could the current results on each language family have been skewed by the higher-resource languages such as English, Spanish, Russian, Hindi, and Chinese within their respective families? In other words, would the proposed strategy also benefit less-represented languages, such as Catalan and Kannada, which have fewer tokens?

---

> ### Author Response · Authors · 2024-11-20
>
> **[W1. Downstream performance]** Please refer to the third point in our general response.
>
> ---
> **[W2. Benefits of loss reduction]** We want to clarify that we focus on the problem of multilingual **pretraining**, where the primary objective is to minimize test cross-entropy loss. The absolute magnitude of cross-entropy loss has limited interpretability, making it challenging to judge whether a specific reduction is "significant" solely based on its size. As observed in prior works [1], the scope for improvement in cross-entropy loss is naturally constrained, which highlights the importance of even small, consistent reductions.
>
> The key strength of our approach is its **consistency in achieving the lowest loss** across diverse weighting scenarios and model sizes. This consistency across different settings demonstrates the effectiveness of the proposed data-mixing strategy and ensures systematic improvements in achieving the pretraining objective compared to alternative mixtures.  Moreover, our approach enables predictive ability of losses across language families for larger compute budgets—whether by scaling parameters, data, or both—without the need for extensive training runs. This predictability, prior to this work, is only enjoyed in the monolingual setting. Crucially, our method introduces control over the loss as a function of the sampling ratios, an aspect not addressed in prior work.
>
> Finally, as discussed in the third point of general response, using downstream tasks as an evaluation metric introduces additional sources of variability and task-specific noise. Compared with the test losses, downstream metrics are indirect and unreliable for testing the effectiveness of our data-mixing strategy for pretraining.
>
> [1] On the Pareto Front of Multilingual Neural Machine Translation. Chen et al. 2023.
>
> ---
> **[W2. Results in Table 6]** We apologize for the discrepancy in the individual values for the last two entries in Table 6; they were inputted incorrectly, but the total normalized loss is accurate as reported. Specifically, the normalized losses for Germanic and Sino-Tibetan should be 1.134 and 1.216, respectively. This can be verified by noting that the results in Table 6 are derived from Table 5 values divided by the mono-family losses. For example, the updated values maintain the expected ratios, as seen with Sino-Tibetan: 1.714/1.216 (Uniform) = 1.759/1.248 (By tokens) = 1.743/1.237 ($\alpha=0.5$) = 1.409, which matches the mono-family loss for Sino-Tibetan. The original value of 1.134 led to an incorrect ratio (1.714/1.134 = 1.511), confirming this as an input error. Other results can be verified in a similar manner. We have updated the manuscript to correct this discrepancy, and we confirm that the overall conclusions and benefits of the proposed sampling strategy remain valid and consistent across model sizes.
>
> ---
> **[W3. Language family hypothesis]** Please refer to the first point in our general response, where we point out the distinctions between cross-family and cross-lingual transfer, and add two additional family combinations to further validate our hypothesis.
>
> ---
> **[W3. Alternative grouping strategies]** Please refer to the second point in our general response.
>
> ---
> **[Q1. Comparison with previous works]** To the best of our knowledge, the only prior work attempting to relate losses with model size and sampling ratios for multilingual model is [1], which focuses exclusively on neural machine translation (NMT) in bilingual settings. At the end of Section 3.4, we provide a detailed comparison with it. In short, our scaling law generalizes and addresses key limitations in [1]. We reiterate some key advancements compared with previous works here:
> - **Explicit modeling of sampling ratios**: In terms of the functional form, we explicitly model the relationship with sampling ratios $p$ by the form $\frac{A_i}{N^\alpha_i}p_i^{-\gamma}$ instead of $\frac{\beta_{p,i}}{N^\alpha_i}$ in [1], where $\beta_{p,i}$ is dependent on $p$ in an **undefined** manner. This explicit formulation enables our scaling law to be predictive for unseen sampling ratios, addressing a key limitation in [1]. In contrast, fitting for $\beta_{p,i}$ in [1] requires surrogate measures and additional heuristics.
> - **Incorporation of dataset size**: We introduce the dependency on dataset size $D$ to provide a holistic understanding of multilingual training dynamics.
> - **Broader applicability**: Our work extends beyond bilingual NMT to general-purpose multilingual language modeling, accommodating multiple language families. This broader scope makes our framework more versatile and suitable for a wide range of multilingual tasks.
>
> [1] Scaling laws for multilingual neural machine translation. Fernandes et al. 2023.

---

> ### Author Response · Authors · 2024-11-20
>
> **[Q2. OOD performance]** We believe separate OOD testing is not suitable for our scaling law, as our pretraining data mixture (CC-100 dataset) already encompasses an extensive and diverse corpus of web crawled content. Given this broad coverage, defining a clear boundary for what constitutes “out-of-distribution” data becomes challenging. In line with common practice in the scaling law literature [1,2,3,4], our evaluation is based on test losses from held-out, in-distribution data to conduct fitting and further analysis.
>
> [1] Scaling Laws for Neural Language Models. Kaplan et al. 2020.
>
> [2] Training Compute-Optimal Large Language Models. Hoffman et al. 2022.
>
> [3] Scaling Laws for Neural Machine Translation. Ghorbani et al. 2021.
>
> [4] Scaling Laws for Multilingual Neural Machine Translation. Fernandes et al. 2023.
>
> ---
> **[Q3. Minority language families]** The Niger-Congo family has very limited data, and due to this data limitation, it does not meet our data sufficiency requirement (line 189). Therefore, the proposed scaling law may not be directly applicable in this case. In fact, in the scaling law literature, extremely low-resource language settings often require specific adjustments and modeling [1,2], which is beyond the scope of this paper and left for future works.
>
> [1] Scaling Data-Constrained Language Models. Muennighoff et al. 2023.
>
> [2] On the Pareto Front of Multilingual Neural Machine Translation. Chen et al. 2023.
>
> ---
> **[Q3. Minority language within families]** To investigate whether higher-resource languages skew the results within each family, we examine individual losses within the Romance family by comparing Spanish (the highest-resource language, with 28.1% of Romance tokens) and Catalan (the lowest-resource language, with 7.5%). We measure their individual losses across different Romance sampling ratios. From the following table, both languages show a consistent decrease in loss as the Romance ratio increases, indicating that each benefits from a higher family sampling ratio.
> |     Romance ratio    |     0.25     |     0.375    |     0.625    |     0.875    |     1        |
> |----------------------|--------------|--------------|--------------|--------------|--------------|
> |     Catalan loss     |     2.759    |     2.675    |     2.546    |     2.468    |     2.421    |
> |     Spanish loss     |     3.008    |     2.931    |     2.813    |     2.740    |     2.695    |
>
> We also fit a power-law relationship (as in Equation (2)) to the individual language losses, with the following results:
> |                |     $\mathcal{L}^\star$    |     $\gamma$     |     R-squared    |
> |----------------|----------------------------|-------------------------------|------------------|
> |     Spanish    |     0.896                  |     0.078                     |     0.9952       |
> |     Catalan    |     0.646                  |     0.094                     |     0.9955       |
>
> The high R-squared values confirm that each language’s loss follows the power-law relationship well. Notably, Catalan, as a lower-resource language, has a smaller $\mathcal{L}^\star$ due to its smaller tokenizer vocabulary, consistent with our findings in Table 2. Its larger decay rate $\gamma$ suggests that it benefits more from an increased family ratio, similar to the pattern we observe for other low-resource families (e.g., Indic). Overall, these results indicate that our approach benefits both high- and low-resource languages within each family, and our reported results accurately reflect the overall performance without sacrificing the gains for less-represented languages.

---

> ### Comment · Reviewer_ig2E · 2024-11-23
> **Response to the Authors' Comments**
>
> Thank you for the authors' response and hard work to perform more experiments to validate their arguments (notably for the proof of the hypothesis and Q3).
>
> **[W1 + W2]** Thank you for your detailed response. I appreciate the focus on pretraining test losses and the comparison with prior scaling law studies [3,4,5]. I also acknowledge the challenges introduced by variability and task-specific noise, as highlighted in [6]. However, "general-purpose multilingual models" naturally raise questions about their downstream utility. As the authors note, there is no consensus on how cross-entropy loss translates to downstream task performance, so for the sake of interpretation and comparison, I think including even a small downstream evaluation would be somewhat nice, similar to practices in related work, such as scaling laws in machine translation [8]. While the authors mention in Q1 and their response to reviewer j6WP that comparing directly with [8] may not make sense, it would be nice if a minimal downstream task evaluation can be proposed as this will provide a better practical applicability of the proposed scaling law.
>
> **[W3]** Thank you for providing more experiments and the table. I have looked over Appendix C, but the graph in Figure 6 seems to only present 2 languages in both axes. Could you please clarify how to interpret the graph considering it is supposed to represent 3 languages?
>
> The discussion of significance in loss reductions appears inconsistent. For instance, while differences in sampling ratios are deemed 'insignificant', in other contexts, a reduction of 0.017 (in the result) is considered meaningful elsewhere. As the authors brought up as well, loss is very limited in interpretation, and it is unclear to me how significant such a difference in loss is. I think statistical analysis may help provide a more meaningful comparison.
>
> ## References
>
> [1] Same Pre-training Loss, Better Downstream: Implicit Bias Matters for Language Models. Liu et al. 2022.
>
> [2] Understanding the Role of Cross-Entropy Loss in Fairly Evaluating Large Language Model-based Recommendation. Xu et al. 2024.
>
> [3] Scaling Laws for Neural Language Models. Kaplan et al. 2020.
>
> [4] Scaling Laws for Autoregressive Generative Modeling. Henighan et al. 2020.
>
> [5] Scaling Laws for Fine-Grained Mixture of Experts. Krajewski et al. 2024.
>
> [6] Are Emergent Abilities of Large Language Models a Mirage? Schaeffer et al. 2023.
>
> [7] Scaling Laws for Predicting Downstream Performance in LLMs. Chen et al. 2024.
>
> [8] Scaling laws for multilingual neural machine translation. Fernandes et al. 2023.

---

> > ### Author Response · Authors · 2024-11-24
> >
> > We want to thank the reviewer for acknowledging our efforts to validate our arguments, and we are glad that our responses have addressed many of the concerns. We hope to further clarify the remaining points:
> >
> > ---
> > **[W1+W2]** We evaluate the 1.2B models from Table 6 on the multilingual MMLU benchmark [1]. Multilingual MMLU is one of the few downstream tasks that support evaluation across all 23 languages in our study. The results, shown in the table below, indicate that the MMLU accuracy somewhat aligns with the test cross-entropy losses, where our proposed method achieves the highest overall accuracy. While the 1.2B models perform only slightly better than random guessing, this is expected because these models are pretrained but not instruction-tuned. This observation is consistent with the findings in the MMLU paper [2], where even GPT-3 small (2.7B parameters) achieves an average accuracy of only 25.9%.
> >
> > |                  |     Romance    |     Slavic    |     Indic    |     Germanic    |     Sino    |     Average    |
> > |------------------|----------------|---------------|--------------|-----------------|-------------|----------------|
> > |     Uniform      |           25.5 |          25.5 |         25.0 |            25.5 |        25.5 |           25.4 |
> > |     By tokens    |           25.3 |          25.7 |         24.8 |            25.5 |        25.3 |           25.3 |
> > |     $\alpha=0.5$    |           25.4 |          25.6 |         24.8 |            25.3 |        25.4 |           25.3 |
> > |     Ours |           25.6 |          25.4 |         25.6 |            25.5 |        25.6 |           25.6 |
> >
> > Nevertheless, while we include this downstream evaluation as requested, **we do not think there is a clear connection between the pretraining quality and specific downstream metrics**. The MMLU results offer some validation of our method's practical benefits, but they also introduce variability from task-specific nuances, which are irrelevant to the effectiveness of our data-mixing strategy for pretraining. In contrast, test losses provide a faithful and direct assessment of the pretraining process, as they are not influenced by downstream task-specific factors.
> >
> > [1] Okapi: Instruction-tuned Large Language Models in Multiple Languages with Reinforcement Learning from Human Feedback. Lai et al. 2023.
> >
> > [2] Measuring Massive Multitask Language Understanding. Hendrycks et al. 2021.
> >
> > ---
> > **[W3. Figure interpretation]** We clarify this by taking Figure 6 (left) as an example. For the experiments of fixed Romance ratio with 0.2 (blue lines), we have the three ratio combinations: {Romance=0.2, Indic=0.2, Slavic=0.6}, {Romance=0.2, Indic=0.4, Slavic=0.4}, {Romance=0.2, Indic=0.6, Slavic=0.2}. Since the Indic and Slavic ratios always add up to 0.8 in this setting, we choose to represent only the Indic ratio on the x-axis for simplicity. This means that each point on the x-axis uniquely identifies a specific combination of the three language ratios. For instance, the Indic ratio of 0.2 corresponds to the combination {Romance=0.2, Indic=0.2, Slavic=0.6}. We hope this explanation helps clarify how to interpret the graph.
> >
> > ---
> > **[W3. Loss significance]** To clarify, in our experiments for hypothesis testing, the normalized losses (averaged over three runs) for the three ratio combinations differ by approximately 0.001 (as shown in Figure 2, left). This difference is an order of magnitude smaller than the 0.017 improvement. Thus, while the magnitude of cross-entropy loss has limited interpretability, we believe this improvement is noteworthy because the observed difference is unlikely to be attributed to randomness. We also want to emphasize that the key strength of our approach is its **consistency** in achieving the lowest loss across diverse weighting scenarios and model sizes, rather than specific absolute reduction in loss.

---

> > > ### Comment · Reviewer_ig2E · 2024-11-24
> > >
> > > Thank you again for the authors' responses.
> > >
> > > **[W1 + W2]** Thank you for your willingness to run an experiment on a downstream task.
> > >
> > > **[W3]** Thank you for modifying the plots in Figure 6, I think they are much clearer now. However, a small thing, I think the placement of the legend for the right plot of Figure 6 can be modified to be consistent with the left and middle plots. Thank you also for your clarification.
> > >
> > > Regarding loss significance, I believe the difference the maximum difference in Romance loss based on the table you have provided in the General Response you provided is 0.006 (2.607 from Romance:Germanic:Slavic ratio of 0.5:0.25:0.25 and 2.601 from Romance:Germanic:Slavic ratio of 0.5:0.375:0.125). But my biggest concern is the orange line in Figure 6 on the right, where there is a huge jump for Sino-Tibetan loss from Sino-Tibetan:Germanic:Slavic ratio of 0.5:0.125:0.375 to Sino-Tibetan:Germanic:Slavic ratio of 0.5:0.375:0.125, which should be more than 0.015 by eye-balling it. They are also consistent based on the standard deviation shade. If 0.017 were a significant improvement, wouldn't this difference be significant, undermining Hypothesis 1?
> > >
> > > Thank you again for your responses thus far, I have adjusted my score accordingly.

---

> > > > ### Author Response · Authors · 2024-11-25
> > > >
> > > > We sincerely thank the reviewer for their engagement with our work, and we are delighted to hear that our response has addressed the majority of the concerns and clarified key aspects of our study. To further address the remaining point, we provide the following response:
> > > >
> > > > **[W3. Loss significance]** We would like to reiterate that **we do not claim the “significance” of any specific value of the losses in isolation**, as we have consistently stated that the absolute magnitude of the loss itself does not provide meaningful interpretation. Instead, we emphasize focusing on the **consistency** of the results and the **overall trends** in the hypothesis testing. The purpose of our hypothesis testing is to demonstrate that language families exhibit minimal cross-family transfer.
> > > >
> > > > To illustrate this, we provide a counterexample in Figure 2 (right), showing what happens when this hypothesis does not hold true for random groupings. In this case, systematic cross-family transfer is manifested by monotonic decrease in loss with changes in sampling ratios of other families. In Figure 6 (right), while it may seem that the Sino-Tibetan loss exhibits a notable change, the observed fluctuations do not exhibit a systematic monotonic pattern associated with cross-family transfer. This suggests that the variation is attributable to randomness rather than cross-family transfer, as corroborated by the consistent behavior across other family combinations.
> > > >
> > > > For the validation of Hypothesis 1, we conducted thorough experiments on 4 family combinations, each involving 6 different sampling ratio configurations. For each of these 24 data points, we report the average results across three runs, resulting in a total of 72 pretraining runs for 85M models on 50B tokens. This is a computationally intensive endeaveor to rigorously validate the hypothesis. Across all these data points, we observe a consistent trend: losses for a fixed language family remain minimally influenced by the sampling ratios of other jointly trained families.
> > > >
> > > > Since our hypothesis can only be assessed empirically, we believe the extensive experiments and consistent results presented provide strong evidence to support its validity. While minor individual variations may occur, they do not undermine the broader trends that align with our hypothesis.

---

> > > > > ### Comment · Reviewer_ig2E · 2024-11-26
> > > > >
> > > > > Thank you again for the authors' responses. I will keep my score as it is as I had already adjusted my score.

---

> > > > > > ### Author Response · Authors · 2024-11-26
> > > > > >
> > > > > > We are glad to hear that all concerns have been addressed, and we sincerely appreciate your valuable comments, which have improved the quality of our paper. We kindly ask if there are any remaining reservations that prevent raising the score to a level of acceptance. If so, we would be more than willing to address them further. Your feedback is immensely valuable to us, and we deeply thank you for the time and effort you have dedicated to reviewing our work.

---

> ### Author Response · Authors · 2024-12-02
>
> Dear reviewer ig2E,
>
> We sincerely appreciate the valuable insights you have provided in the review, and we have diligently worked to address your comments through our rebuttal. As a kind reminder, **tomorrow is the last day that the reviewer can post messages to authors**. Since all concerns have been addressed, please let us know if there are any remaining reservations that prevent raising the score to a level of acceptance. If so, we would be more than willing to address them further.
>
> Thank you again for your time and effort!
>
> Authors

---

### Official Review · Reviewer_j6WP · 2024-11-02

**Soundness:** 2
**Presentation:** 3
**Contribution:** 2
**Rating:** 6
**Confidence:** 3

**Summary:**

This paper introduces a method to calculate the multilingual scaling law and estimate the optimal sampling ratio for each language during multilingual pre-training. Experimental results on models with 85M and 1.2B parameters show that the optimal sampling ratios estimated are better than the three naive baselines.

**Strengths:**

- The scaling law proposed is more fit for multilingual pre-training by considering cross-lingual transfer.
- There is an interesting finding that $\gamma_{i}(N, D)$ is independence from N and D.
- The optimal sampling ratios inferred can bring marginal improvement (< 0.1) than the three naive baselines.

**Weaknesses:**

- **Missing important baselines**: The three baselines in this paper are naive. More baseline methods like the Equation (7) in Fernandes et al., 2023 are not incorporated.

- **More experiments needed**: The minimal cross-group transfer hypothesis (hypothesis 1) is hard to quantify and meet. It is still far from the real phenomenon. To support hypothesis 1, experiments are only conducted on (Romance, Indic) and (Sino-Tibetan, Slavic) language family pairs with varying three sampling ratios in {0.2, 0.4, 0.6}. How about the sampling ratio for other language family pairs varies? (For example, the results for more similar language families like Germanic and Romance or Romance and Slavic)

- **Overstatements**:
  1. The first sentence in the abstract "We propose a novel scaling law for general-purpose decoder-only language models(LMs) trained on multilingual data, **addressing the problem** of balancing languages during multilingual pretraining." Actually, the problem is simplified into 5 language families and cases of low-resource languages are reduced in this paper (Line 190-191).

  2. The lines in Figure 4 are nearly parallel, and it is better to report the slope rate for each line.

- **Typos**:
  1. The x-y axis for Figure 4: $L(p)-p$ --> $log(L(p))-log(p)$
  2. Line 452: $q_j$ --> $q_i$
  3. Line 515: asses --> assess

**Questions:**

a. (Hypothesis 1) How about the sampling ratio for other language family pairs varies? For example, the results for more similar language families like Germanic and Romance or Romance and Slavic.

b. As shown in Table 5, it can be found that the probabilities of Romance, Germanic, and Sino-Tibetan differ between models with 85M or 1.2B parameters. How to interpret this phenomenon?

c. (Line 531, 100 models) Does it refer to the number of models investigated?

---

> ### Author Response · Authors · 2024-11-20
>
> **[Missing baselines]** To clarify, [1] models the scaling laws for each **discrete** value of $\beta_{p,i}$, which in turn **cannot** be used for predicting the optimal sampling value. In contrast, we demonstrate that the scalar constants obtained in their work actually follow a continuous power law as a function of $p$. This continuous formulation of $p$ allows us to get an optimal sampling distribution, which is one of the fundamental contributions of this work. In addition, the study in [1] is limited to bilingual settings, and this limitation means that Equation (7) cannot be applied to a true multilingual setting with more than 2 languages.
>
> [1] Scaling laws for multilingual neural machine translation. Fernandes et al. 2023.
>
> ---
> **[More experiments needed]** Please refer to the first point in our general response, where we add two additional family combinations to further validate our hypothesis.
>
> ---
> **[Overstatements]** We agree with the reviewer’s suggestion, and we have updated the wording in the manuscript to better reflect our contribution.
>
> ---
> **[Slope in Figure 4]** We report the slopes in the following tables. The values are close among different lines, validating our claims.
> |     N       |     Slope     |
> |-------------|---------------|
> |     85M     |     -0.080    |
> |     397M    |     -0.080    |
> |     810M    |     -0.076    |
> |     1.2B    |     -0.074    |
>
> |     D       |     Slope     |
> |-------------|---------------|
> |     25B     |     -0.088    |
> |     50B     |     -0.085    |
> |     100B    |     -0.083    |
>
> ---
> **[Ratio difference between 85M and 1.2B]** The difference in sampling ratios between the 85M and 1.2B models primarily reflects variations in the benefits that each language family receives from scaling up the model size. During optimization (Equation (8)), the values of $\mathcal{L_i}^\star$ differ between 85M and 1.2B models, leading to slight adjustments in the optimal sampling ratios. Empirically, we observe that scaling up the model size provides a greater reduction in loss for the Sino-Tibetan family compared to Romance, which explains why the Sino-Tibetan ratio increases slightly with the 1.2B model. However, this shift is minor and does not significantly impact final performance, as demonstrated by the minimal difference of 0.003 in those ratios.
>
> ---
> **[100 models]** 100+ models refers to the number of experiments we ran spanning different sampling ratios, model sizes and dataset sizes to propose and empirically validate the scaling laws spanning all the three scaling dimensions.

---

> ### Author Response · Authors · 2024-11-25
>
> Dear reviewer j6WP,
>
> We sincerely appreciate the valuable insights you have provided in the review, and we have diligently worked to address your comments through our rebuttal. As a kind reminder, **the author-review discussion period will end tomorrow**. Should you have any further inquiries or seek additional clarifications, please do not hesitate to let us know. If the questions have been resolved, we would greatly appreciate it if you could acknowledge our response and consider updating the score.
>
> Thank you again for your time and effort!
>
> Authors

---

> ### Comment · Reviewer_j6WP · 2024-11-26
> **Response to Authors**
>
> Thank you for your responses. My concerns have been addressed, and reviews are updated correspondingly.

---

> > ### Author Response · Authors · 2024-11-26
> >
> > We are glad to hear that all concerns have been addressed, and we sincerely appreciate your valuable comments.

---

### Official Review · Reviewer_ojYU · 2024-11-09

**Soundness:** 3
**Presentation:** 3
**Contribution:** 3
**Rating:** 5
**Confidence:** 2

**Summary:**

This paper looks at the oft-neglected area of multilingual language model and in particular, scaling laws. They note that most prior work in scaling laws is focused on monolingual models or translation models (as opposed to broader generation models). The authors train more than 100 models varying in size from 85 million parameters to 1.2 billion (not counting embeddings) for 5 language families. Overall, they claim their sampling strategy is better at getting overall better multilingual loss.

**Strengths:**

An overlooked area in the field. Multilingual Language models are very important.

Lots of experiments.

**Weaknesses:**

A bit dense (especially some captions of figures and tables that took me a while to figure out.

For instance my thoughts about "Figure 1 Model size N figure is a bit confusing to me. What is N? 0,2,4,8 of what? Oh, 8 is 85Million parameters. It could be a bit clearer in the caption."

Figure 5 left and middle … a bit hard to know. Y-axis is loss (for a sampling ratio?) X-axis is sampling ratio? Colored lines are model sizes. Data size is just difference between left and middle graph? The caption should really be more explicit. Also, what am I supposed to take away from this?

**Questions:**

Does better loss generalize to better downstream performance on other tasks? I would assume so I was just curious if you had tried anything?

Figure 1 appears to show that all models get better with more data, larger model size, and sampling ratio of 1. Why use a uniform sampling rate of 0.2 and not just 1.0? I assume that 1.0 implies 0.0 of others meaning 0.2 means equal sampling of all. If so, do you use all the data completely? And does this outperform other baselines? My guess is that this is shown somewhere in tables 3-6 but it is unclear to me.

What happens if you randomly group languages (not by family) and apply your scaling laws? Do the same conclusions hold? I'm trying to get at your claims that language family matter more - but I don't think you test languages apart from their language family ever.

I’d argue that MT is a type of generation and cache your claims in the intro a bit more.

---

> ### Author Response · Authors · 2024-11-20
>
> **[Figure 1]** We have updated the caption of Figure 1 to explicitly point out that $N$ refers to the number of parameters.  In addition, the legend at the bottom of the x axis specifies the unit of the parameters (1e8).
>
> ---
> **[Figure 5]** As detailed in the caption, Figure 5 shows the fitted law on 50B and 100B tokens. The main takeaway is that the scaling law across sampling ratios holds for different model sizes as well as different dataset sizes, as the empirical results (points) lie closely on the fitted curves (dashed curves).
>
> ---
> **[Downstream performance]** Please refer to the third point in our general response.
>
> ---
> **[Sampling ratios]** We define the sampling ratios as the proportion of a language family within the entire training mixture, so the ratios must sum up to 1. While it is true that a language family performs optimally when all training tokens are dedicated to it alone (i.e., a sampling ratio of 1.0 for that family and 0.0 for others), this scenario limits the model’s multilingual capabilities. Our objective is to study the effects of different data allocations within a fixed token count budget (50B, 100B, and 200B tokens), which are subsets of the full CommonCrawl data.
>
> The strategy of using 0.2 for all families is referred to as uniform sampling in Table 3 through 6. Our optimization approach, which determines optimal sampling ratios, consistently outperforms this uniform baseline across the different token count budgets.
>
> ---
> **[Random grouping]** We have demonstrated this result in the right figure of Figure 2. In short, we find that there exists significant cross-group transfer in the randomly grouped scenario because languages from the same family are split into separate groups. This demonstrates the effectiveness of family-based grouping. Please refer to Section 3.2 for more details.

---

> > ### Comment · Reviewer_ojYU · 2024-12-03
> > **Romance and Germanic language families quite similar**
> >
> > Thanks for running the experiments in Figure 2. However, I think there are significant confounding factors in this analysis. First, Romance and Germanic languages are relatively closely related. Second, using group 1 and group 2 languages with Indic languages (which I assume to mean different scripts) adds another variable that makes it difficult to disentangle the interaction therein.

---

> ### Author Response · Authors · 2024-11-25
>
> Dear reviewer ojYU,
>
> We sincerely appreciate the valuable insights you have provided in the review, and we have diligently worked to address your comments through our rebuttal. As a kind reminder, **the author-review discussion period will end tomorrow**. Should you have any further inquiries or seek additional clarifications, please do not hesitate to let us know. If the questions have been resolved, we would greatly appreciate it if you could acknowledge our response and consider updating the score.
>
> Thank you again for your time and effort!
>
> Authors

---

> ### Author Response · Authors · 2024-12-02
>
> Dear reviewer ojYU,
>
> We sincerely appreciate the valuable insights you have provided in the review, and we have diligently worked to address your comments through our rebuttal. As a kind reminder, **tomorrow is the last day that the reviewer can post messages to authors**. Please do not hesitate to let us know if you have further inquiries. If the questions have been resolved, we would greatly appreciate it if you could acknowledge our response and consider updating the score.
>
> Thank you again for your time and effort!
>
> Authors

---

> ### Author Response · Authors · 2024-12-03
>
> Thank you for engaging with the discussion.
>
> We believe the reviewer may have misunderstood the purpose of Figure 2. The aim of this analysis is not to study the degree of similarity between families, but rather to demonstrate that **cross-family transfer is minimal**. This claim holds true **regardless of similarities between families**, as shown by the consistent results observed across four different family combinations (the other three in Figure 6). While it is reasonable to assume some degree of relatedness among **individual languages** in these families, our experiments show that this relatedness does not lead to noticeable cross-family transfer. The key intuition here is that each language family already has sufficient internal data, reducing its reliance on other families for performance improvements. In fact, this family-based analysis is a key contribution of our work, distinguishing it from traditional analysis based on individual languages.
>
> Regarding the random grouping experiments, the inclusion of Indic language refers to dividing Indic languages across multiple groups (e.g., Hindi in Group 1 and Kannada in Group 2). In Figure 2 (right), we can see that when languages from the same family is distributed across two groups, significant cross-group interaction occurs. This highlights that random grouping fails to meet the criteria outlined for our hypothesis (lines 188–190), reinforcing the importance of family-based grouping in ensuring minimal cross-group transfer.
>
> We hope this resolves the misunderstandings and clarifies the design and intent of our analysis.

---

### Author Response · Authors · 2024-11-20

We want to thank all the reviewers for their insightful feedback and the time invested in reviewing our paper. We are particularly grateful for the acknowledgement of several key aspects of our work:
- We study an important yet underrepresented problem in multilingual scaling law (Reviewer ojYU, Czqd).
- Our language-family-based setup simplifies the complexity of cross-lingual interactions during training, enhancing scalability across multiple languages (Reviewer ig2E, j6WP).
- Our proposed scaling law provides an effective way to optimize for the multilingual training data mixture, providing practical guidance for data allocation (Reviewer j6WP, ig2E, Czqd).

---

We hope that the following response can address the reviewers’ concerns:

**[More proof of hypothesis]** We have conducted additional experiments to further validate our hypothesis across more closely related language families as suggested by the reviewers. Using the same testing procedure from Section 3.2, we experiment with combinations of Romance, Germanic, and Slavic families. Since the reviewers note that high-resource languages like English (within Germanic) may transfer well to related languages in the Romance family (e.g., Spanish, French), we test settings where the Romance sampling ratio is fixed at either 0.2 or 0.5, while varying the proportions of Germanic and Slavic. The results, summarized in the table below, show that changes in the Germanic sampling ratio do not significantly impact the Romance loss either. This supports our hypothesis that the performance of each language family is primarily influenced by its own sampling ratio, with minimal dependence on inter-family interactions. We observe the same pattern when fixing Germanic ratios and vary Romance ratios.

|     Romance ratio    |     Germanic   ratio    |     Slavic ratio    |     Romance loss    |
|----------------------|-------------------------|---------------------|---------------------|
|     0.2              |     0.2                 |     0.6             |     2.782           |
|     0.2              |     0.4                 |     0.4             |     2.784           |
|     0.2              |     0.6                 |     0.2             |     2.787           |
|     0.5              |     0.125               |     0.375           |     2.606           |
|     0.5              |     0.25                |     0.25            |     2.607           |
|     0.5              |     0.375               |     0.125           |     2.601           |

The results highlight the distinctions in the analysis of transfers among **language families** compared with **individual languages**. It demonstrates that the language-family-based approach allows for sufficient intra-family transfer, reducing reliance on inter-family interactions. In fact, this is a key insight and contribution of our work. The results and explanations are updated in both Section 3.2 (Figure 2 (left), line 198-209) and Appendix C (Figure 6, line 695-700) in our revised manuscript.

---
**[Language grouping strategy]** We do not claim that grouping by language families is the only viable or optimal approach for all contexts, as noted in footnote 3 on page 4, where we suggest alternative grouping strategies as well. Instead, we define two key requirements for effective groupings: minimal cross-group transfer and data sufficiency. Among all the grouping methods, we find family-based grouping is a natural choice that aligns with our two requirements, and empirically we verify that this hypothesis holds well. The additional considerations brought up by the reviewers would be valuable for future explorations into grouping strategies that incorporate areal or typological factors.

---

> ### Author Response · Authors · 2024-11-20
>
> **[Downstream performance]** Analogous to prior scaling law works [3,4,5], we focus on the scaling behaviors during the **pretraining** stage, where the objective is to minimize test cross-entropy loss. Consequently, our evaluation centers around test losses, which directly measure the effectiveness of the pretraining process in achieving this goal. In contrast, there is no common consensus on how test losses translate to downstream performance, and this problem alone is an active field of study [1,2,7]. It is highly likely that a better-pretrained model might perform better on certain downstream tasks but worse on others, making it challenging to draw definitive conclusions from downstream evaluations.
>
> In addition, for general language modeling, it is a well-established practice to fit scaling laws based on test losses instead of downstream metrics [3,4,5]. A key reason is the discontinuity and limited resolution of the evaluation metrics (e.g., accuracy), which causes downstream performance to often deviate from smooth and predictable trends, leading to the so-called “emergence” phenomenon [6]. This unpredictability makes downstream evaluation less reliable for reflecting the effects of different pretraining mixtures.
>
> While downstream performance is not the primary focus of our study, we believe our proposed scaling law provides foundational understanding necessary to guide improvements in multilingual pretraining. Developing an accurate law for predicting downstream performance from losses, especially with the added complexity of multilinguality, is an important and challenging problem. While it is beyond the scope of this work, it presents a promising direction for future research building upon the results presented in this paper.
>
> [1] Same Pre-training Loss, Better Downstream: Implicit Bias Matters for Language Models. Liu et al. 2022.
>
> [2] Understanding the Role of Cross-Entropy Loss in Fairly Evaluating Large Language Model-based Recommendation. Xu et al. 2024.
>
> [3] Scaling Laws for Neural Language Models. Kaplan et al. 2020.
>
> [4] Scaling Laws for Autoregressive Generative Modeling. Henighan et al. 2020.
>
> [5] Scaling Laws for Fine-Grained Mixture of Experts. Krajewski et al. 2024.
>
> [6] Are Emergent Abilities of Large Language Models a Mirage? Schaeffer et al. 2023.
>
> [7] Scaling Laws for Predicting Downstream Performance in LLMs. Chen et al. 2024.

---

### Meta-Review · Area_Chair_yKEe · 2024-12-19

**Metareview:**

The paper investigates scaling laws for multilingual language models (MLMs), proposing a method to predict optimal sampling ratios for languages within the training mixture by grouping languages into families. The authors argue that focusing on language families rather than individual languages simplifies cross-lingual transfer analysis and scalability. They claim minimal cross-family transfer and validate these hypotheses using test cross-entropy loss as a measure, conducting experiments across 23 languages from five families with models ranging from 85M to 1.2B parameters. Key contributions include a power-law relationship linking dataset size, model size, and sampling ratios, which generalizes small model findings to larger ones and provides guidelines for multilingual training.

Strengths include addressing an underexplored area in multilingual scaling laws, with rigorous experimentation on diverse language families. The proposed method offers a potentially cost-effective way to optimize data allocation, and the clarity of presentation was noted. However, there are also significant weaknesses: The absence of downstream task evaluations limits the paper's practical implications, and the claimed minimal cross-family transfer is not convincingly demonstrated across diverse settings. Improvements in normalized loss are marginal, questioning the significance of the proposed method’s benefits. Moreover, the reliance on language family grouping overlooks alternative groupings, and the scalability to low-resource languages remains unclear.

Despite revisions addressing some concerns, key limitations persist. The evidence supporting minimal cross-family transfer, a central hypothesis, remains insufficient, as confounding factors (e.g., relatedness within language families) are not adequately addressed. The marginal gains in performance, even when validated, do not substantiate the methodology's utility. Given these issues, the decision is to reject the paper.

**Additional Comments On Reviewer Discussion:**

During the rebuttal period, several key points were raised and discussed by the reviewers. One central critique was the absence of downstream task evaluations to validate the practical applicability of the proposed scaling law. Reviewers, including ig2E, argued that focusing solely on test cross-entropy loss limits the relevance of the findings. The authors acknowledged this concern but defended their choice, stating that downstream metrics introduce task-specific variability and are less reliable for assessing pretraining efficacy. While they conducted a small downstream evaluation (MMLU), it was insufficient to address broader applicability concerns fully.

Another significant issue was the claim of minimal cross-family transfer. j6WP and ig2E questioned the robustness of this hypothesis, pointing out potential confounding factors such as high-resource languages skewing results within families. The authors provided additional experiments and clarified their methodology, but the evidence remained limited to a narrow set of language family pairs, leaving doubts about the generalizability of the hypothesis.

The marginal improvements in normalized loss were also debated. ojYU and ig2E highlighted that the observed gains (e.g., 0.017 reduction) were minimal and questioned their practical significance. The authors argued that even small, consistent reductions are meaningful in the context of pretraining. However, this justification was not universally convincing, particularly given the resource-intensive nature of the experiments.

Lastly, the grouping of languages by families was challenged, with reviewers suggesting alternative groupings based on typological or geographical factors. The authors defended their choice, citing data sufficiency and empirical results, but did not extend their analysis to alternative grouping strategies.

In weighing these points, the lack of downstream task validation carried significant weight in the final decision, as it directly impacts the practical utility of the proposed method. The evidence for minimal cross-family transfer was judged insufficient to substantiate the authors' claims, and the marginal gains in normalized loss did not justify the computational costs of the approach. While the authors made efforts to address reviewers' concerns, the key criticisms remained unresolved.

---

### Decision · Program_Chairs · 2025-01-22

Reject